# ATLAS: Automatic Local Symmetry Discovery

## Abstract

Existing symmetry discovery methods predominantly focus on *global* transformations across the entire system or space, but they fail to consider the symmetries in *local* neighborhoods. This may result in the reported symmetry group being a misrepresentation of the true symmetry. In this paper, we formalize the notion of local symmetry as atlas equivariance. Our proposed pipeline, **au**tomatic **l**ocal **s**ymmetry discovery (`ATLAS`), recovers the local symmetries of a function by training local predictor networks and then learning a Lie group basis to which the predictors are equivariant. We demonstrate `ATLAS` is capable of discovering local symmetry groups with multiple connected components in top-quark tagging and partial differential equation experiments. The discovered local symmetry is shown to be a useful inductive bias that improves the performance of downstream tasks in climate segmentation and vision tasks.

## 1 Introduction

Equivariant neural networks (Bronstein et al., 2021), a family of models that exploit symmetry as an inductive bias for neural network architectures, have received increasing attention in deep learning due to their training efficiency and improved generalization (Krizhevsky et al., 2017; Worrall & Welling, 2019; Zaheer et al., 2017). The key idea behind these models is that many real-world situations exhibit inherent symmetries—transformations such as rotation, translation, and scaling, which leave the essential properties of a system unchanged. This has enabled a wide range of applications, leading to empirical success (Winkels & Cohen, 2018; Brown & Lunter, 2018; Cohen & Welling, 2016; Cohen et al., 2018). Despite its achievements, equivariant networks require knowledge of the system's symmetries beforehand. To adhere to this successful design principle even when the symmetry group is unknown a priori, many works have developed auxiliary neural networks to automatically identify symmetries (Benton et al., 2020; Zhou et al., 2021; Dehmamy et al., 2021; Moskalev et al., 2022; Yang et al., 2023; Gabel et al., 2023).

The aforementioned equivariant models and symmetry discovery pipelines focus on *global* symmetries, where a transformation applies across the entire space. However, arbitrary manifolds generally do not have global symmetries to begin with (Gerken et al., 2023), preventing the use of globally equivariant networks. This begs the need to consider *local* symmetries—transformations on small neighborhoods—which are much more generalized (Figure 1). Indeed, some recent work considers local symmetry, such as Cohen et al. (2019), which develops gauge equivariant CNNs to take advantage of the gauge symmetries of arbitrary manifolds. Construction of such networks once again requires knowledge of the symmetry beforehand. Existing discovery methods may not be applicable as they fail to consider local symmetries. Hence, the

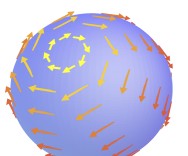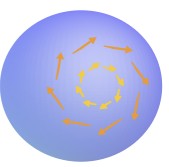

Figure 1: Global vs local transformations. Global transformations (left) alter the space in a uniform manner, whereas local transformations (right) only affect a particular neighborhood.

need to develop a local symmetry discovery pipeline is clear: it would open up symmetry discovery to broader domains, such as partial differential equations or computer vision tasks on arbitrary meshes and manifolds.

In this work, we define local symmetry around the notion of an atlas. An atlas is a collection of local regions, or charts, that cover a manifold. In short, the principle of *atlas equivariance* states

that when restricting a function to a particular chart, the localized function must be equivariant. We develop a method based on deep learning that can discover the atlas equivariances of dataset in the form of a Lie group. To do so, we first model the task function localized to the various charts using neural networks. Then, we create a Lie group basis and optimize it until the localized networks are equivariant with respect to the group. After discovery, we use the resulting symmetry as an inductive bias to create equivariant networks. Specifically, we experiment with top-quark tagging, synthetic partial differential equations, MNIST classification, and climate segmentation to test the validity of our discovery method and measure performance gains in downstream models.

Our contributions can be summarized as follows:

- We formalize the notion of local symmetry through the definition of *atlas equivariance*.

- We develop a pipeline, **au**tomatic **l**ocal **s**ymmetry discovery (`ATLAS`), to recover local symmetries from a dataset. `ATLAS` can learn both continuous and discrete symmetries.

- `ATLAS` can discover local symmetries in cases where existing global symmetry search methods are not applicable.

- We show that incorporating the symmetries discovered by `ATLAS` in a gauge equivariant CNN (Cohen et al., 2019) can lead to better performance and parameter efficiency.

## 2 RELATED WORK

**Equivariant Neural Networks**. Equivariant neural networks use known symmetry as an inductive bias when fitting a model. Group equivariant CNN extends the translational equivariance of a CNN to rotations and reflection using group theory (Cohen & Welling, 2016). Other works focus on designing networks that are equivariant to a wider range of transformations; particularly, $E(2)$ transformations on Euclidean plane (Weiler & Cesa, 2019), rotations on the sphere (Cohen et al., 2018), Lorentz group transformations (Gong et al., 2022), and $E(n)$ transformations in higher-dimensional spaces (Satorras et al., 2021). These works focus on global transformations and require prior knowledge of the symmetry. In contrast, our pipeline focuses on discovering local symmetry.

Gauge equivariant neural networks extend the ideas of globally equivariant neural networks by enforcing local symmetries instead of global ones. Gauge equivariance has been applied in various contexts, including surface meshes (De Haan et al., 2020), lattice structures (Favoni et al., 2022), and general manifolds (Cohen et al., 2019). Once again, these networks require user intervention to determine the gauge group. On the other hand, we show that our definition of local symmetry is connected to gauge equivariance and that using our discovered symmetry in a gauge equivariant CNN can lead to better performance and parameter efficiency.

**Automatic Symmetry Discovery**. Many works perform automatic symmetry discovery to identify the unknown symmetry within a dataset. Attempts have been made to discover general continuous symmetries with Lie theory, such as LieGG to discover symmetry from the polarization matrix (Moskalev et al., 2022), L-conv (Dehmamy et al., 2021) to find group equivariant functions, and Forestano et al. (2023) who discover closed Lie subalgebras from a dataset. LieGAN (Yang et al., 2023) uses a generator-discriminator pattern to discover global symmetries in the form of both continuous Lie groups and discrete subgroups. Gabel et al. (2023) aim to find the symmetry group as well as quantify the exact distribution of transformations present in a dataset. Although we also rely on Lie theory, we search for local symmetries instead of global ones.

Symmetry discovery works also consider slightly varied problem settings. Some authors seek to find a subset of possible symmetries (Benton et al., 2020; Romero & Lohit, 2022). Others consider the case where the group acts on the latent space instead of the feature space (Yang et al., 2024; Gabel et al., 2023; Keurti et al., 2023; Koyama et al., 2023). Still, these papers mainly focus on global transformations. More relevant to local symmetry discovery is the work by Decelle et al. (2019), which attempts to see if two datapoints are related by a particular local transformation. This is distinct from `ATLAS`, where we characterize the local symmetry group in an interpretable manner.

## 3 BACKGROUND

We provide background information on Lie groups, equivariance, feature fields, and atlases. We assume some knowledge of group theory and otherwise refer readers to Artin (2011); Weiler et al. (2021) as useful starting points.

**Lie Groups**. A Lie group is a group that is also a differentiable manifold. Some examples include $SO(2)$, $O(3)$, and $SL(3)$. The Lie algebra of a Lie group, denoted $\mathfrak{g}$, is the tangent space at the identity element. Being a vector space, Lie algebra is often simpler to work with than the group.

For matrix Lie groups, the matrix exponential $\exp(A)$ provides a way to map elements of the Lie algebra to elements of the group's identity component $G_0$, i.e. the connected component containing the identity element. The various connected components are cosets of $G_0$ and will also be plainly referred to as cosets in our work. In many cases, we can factor an arbitrary element of the group as a product, $g = C_i \cdot \exp(A)$, for some coset representative $C_i \in G$ and Lie algebra element $A \in \mathfrak{g}$. Thus, to understand a Lie group, it is often enough to enumerate all the cosets in the component group $G/G_0$, and identify a basis for its Lie algebra. For further information, see Kirillov (2008).

**Equivariance**. A function $f$ is said to be $G$-equivariant for some group $G$ if the following holds:

$$(\forall g \in G) \quad f(g \cdot x) = g \cdot f(x) \tag{1}$$

Here, $g \cdot x$ and $g \cdot f(x)$ denote (possibly different) group actions.

**Feature Fields**. A feature field identifies a feature vector for each point in a manifold $\mathcal{M}$. Specifically, a feature field is given as a map $F : \mathcal{M} \to \mathbb{R}^d$, where $d$ is the dimension of the feature field.

**Charts and Atlases**. It is not possible to give a consistent choice of coordinates across manifolds with non-trivial topology. We define local coordinates in terms of local charts. A chart is a pair $(U, \varphi)$ where $U$ is an open subset of $\mathcal{M}$ and $\varphi$ is a homeomorphism from $U$ to an open subset of Euclidean space. An atlas is a set of charts that collectively cover a manifold $\mathcal{M}$.

## 4 ATLAS: AUTOMATIC LOCAL SYMMETRY DISCOVERY

Despite the achievements of existing symmetry discovery methods, the challenge of local symmetry discovery remains largely untouched. To address this problem, we first formulate atlas equivariance as a definition of local symmetry. Then, we detail our methodology for discovering local symmetry in the form of a Lie algebra basis and component group. Finally, we highlight theoretical connections to existing work as well as implementation notes.

### 4.1 ATLAS EQUIVARIANCE

To provide an intuition of local symmetry, we highlight the heat equation on a torus in Figure 2 as a concrete example. Consider the time-stepping function that evolves the current state of the system for some fixed time interval. If we focus only on a neighborhood of the input feature field and its corresponding neighborhood in the output field, a local rotation in the input results in an identical local rotation to the output.

To define local symmetry formally, assume we have a map $\Phi$ that transforms an input feature field $F_{\text{in}} : \mathcal{M} \to \mathbb{R}^{d_{\text{in}}}$ to an output field $F_{\text{out}} : \mathcal{M} \to \mathbb{R}^{d_{\text{out}}}$. Then, suppose $\mathcal{A}$ is an atlas on $\mathcal{M}$ given by a finite collection of charts $\{(U_c, \varphi_c)\}_{c=1}^N$. We can relocate a feature field on $\mathcal{M}$ restricted to the neighborhood $U_c$ to a flat Euclidean space by pulling back over $\varphi_c^{-1}$:

$$\left((\varphi_c^{-1})^* F\right)(x) = \begin{cases} F(\varphi_c^{-1}(x)) & \text{if } x \in \varphi_c(U_c) \\ 0 & \text{else} \end{cases} \tag{2}$$

Note that the flattened feature field is trivially extended outside $\varphi_c(U_c)$. This is necessary for introducing atlas equivariance, where the group action may take a point $p \in \varphi_c(U_c)$ outside this original domain. This 0-padding can also be replaced with another value appropriate to the context.

We define local (atlas) equivariance for task functions where the output signal depends locally on the input signal. We formalize this as $\mathcal{A}$ locality. The intuitive notion of a local function is that, under

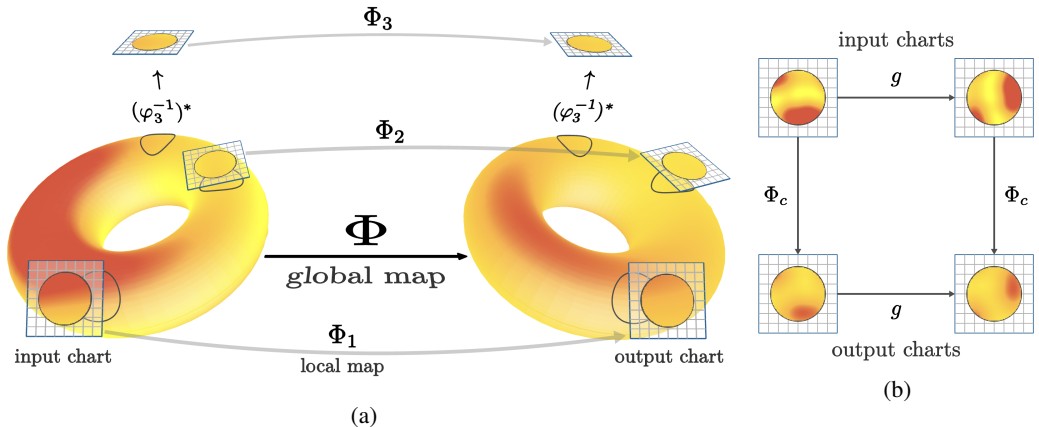

Figure 2: Atlas equivariance explained through the example of the heat equation. (a) highlights how the task function $\Phi$ is a function whose input and output are scalar feature fields on a torus. $\Phi$ is then broken up into localized functions, i.e. the $\Phi_c$. Although we only highlight three $\Phi_c$ for visual purposes, in reality there is one for each chart. (b) is a commutative diagram that highlights the rotational equivariance of a localized function and hence the rotational atlas equivariance of $\Phi$.

an appropriate choice of atlas, we can fully reconstruct the output field in any individual chart $U_c$ solely from the input field along the same chart. Thus, we say $\Phi$ is $\mathcal{A}$ atlas local if we are able to decompose it into various $\Phi_c$, where $\Phi_c$ is a map between the pullback of the $c$th chart of the input and output feature fields. We sometimes refer to $\{\Phi_c\}$ as the localized functions of $\Phi$. Formally,

**Definition 1** (Atlas Locality). $\Phi$ is $\mathcal{A}$ **atlas local** if for each chart $c$ in $\mathcal{A}$ and arbitrary $F : \mathcal{M} \to \mathbb{R}^{d_{in}}$, there exists a $\Phi_c$ such that $\Phi_c\left((\varphi_c^{-1})^* F\right) = (\varphi_c^{-1})^* \Phi(F)$ when restricted to $\varphi_c(U_c)$.

Here, the restriction of the feature field to the subset $\varphi_c(U_c)$ indicates that we are not particular about the output of $\Phi_c$ outside the projected chart. For these atlas local functions, it is possible to consider the symmetry transformations that operate within local neighborhoods. We formalize this notion of local symmetry as follows.

**Definition 2** (Atlas Equivariance). $\Phi$ is $\mathcal{A}$ **atlas equivariant** to some group $G$ if $\Phi$ is $\mathcal{A}$ atlas local with localized functions $\{\Phi_c\}$ and all $\Phi_c$ are globally $G$ equivariant. Specifically, for the group action $(g \cdot E)(p) = E(g^{-1}p)$ where $E$ is a feature field on the Euclidean space, we must have $\Phi_c(g \cdot (\varphi_c^{-1*}F)) = g \cdot \Phi_c(\varphi_c^{-1*}F)$ for arbitrary $g \in G$ and feature fields $F : \mathcal{M} \to \mathbb{R}^{d_{in}}$.

A technical note is that the $\Phi_c$ may not be unique in that for a given chart $c$, there are many potential localized maps that satisfy the condition specified in Definition 1. Therefore, $\Phi$ is said to be atlas equivariant if, for each chart $c$, any potential $\Phi_c$ is $G$ globally equivariant.

## 4.2 ATLAS EQUIVARIANCE DISCOVERY

In our problem setup, we assume we have a dataset $\{(X_i, Y_i)\}_{i=1}^{n} \subset \mathcal{X} \times \mathcal{Y}$ that models some unknown function $\Phi : \mathcal{X} \to \mathcal{Y}$ between feature fields on the same manifold. We further assume that a suitable atlas $\mathcal{A}$ for the problem is known. We aim to find the maximal matrix Lie group that $\Phi$ is $\mathcal{A}$ atlas equivariant to.

We propose `ATLAS`, an algorithm for **aut**omated **loca**l **s**ymmetry discovery to tackle the problem. First, we must model individual localized functions with neural networks or other differentiable oracles. The exact method is unique to each task, but is generally a simple regression problem. Further details are available in Appendix C. In contrast to global discovery techniques, we emphasize that the individual neural networks are localized maps rather than functions over the entire manifold.

We then find the equivariance group of the localized functions. There are important differences between our procedure and global symmetry discovery. In our setting, a group element must only act on a local chart, rather than the full space. Likewise, a core tenet of local symmetry is that we can

apply different transformations to different regions of the feature field. Care must be taken, therefore, that the actions on different charts are truly independent.

We aim to discover the maximal group of local symmetry. In practice, these symmetries often involve both discrete and continuous transformations, which motivates us to use Lie groups to describe the symmetries of interest. Specifically, we seek to discover both the **Lie algebra** which characterizes the continuous transformations, and the **cosets** that describe the discrete actions of the target group. Put together, these allow us to describe a wide variety of Lie groups. Algorithm 1 outlines our overall procedure, with the details of the subroutines introduced in the following subsections. We analyze the time and space complexity of ATLAS in Appendix E.

---

**Algorithm 1** ATLAS: Automatic Local Symmetry Discovery

---

**input** Dataset $\mathcal{D} = \left\{ \left( X_i : \mathcal{M} \to \mathbb{R}^{d_{\text{in}}}, Y_i : \mathcal{M} \to \mathbb{R}^{d_{\text{out}}} \right) \right\}_{i=1}^n$, atlas $\mathcal{A} = \{(U_c, \varphi_c)\}_{c=1}^N$
**output** Lie algebra $\{B_i\}$, cosets $\{C_\ell\}$
    Train a predictor network $\Phi_c$ for each chart $(U_c, \varphi_c)$ under the loss $\mathcal{L}\left( \Phi_c((\varphi_c^{-1})^* X_i), (\varphi_c^{-1})^* Y_i \right)$
    Given the trained predictors $\{\Phi_c\}$, discover the Lie algebra basis $\{B_i\}$    {Section 4.2.1}
    Given the trained predictors $\{\Phi_c\}$ and the discovered Lie algebra basis $\{B_i\}$, find the coset representatives $\{C_\ell\}$    {Section 4.2.2}
    **Return** $\{B_i\}, \{C_\ell\}$

---

### 4.2.1 DISCOVERING INFINITESIMAL GENERATORS

To discover the Lie algebra, we view it as a vector space and learn its basis, also known as the infinitesimal generators (of the group). To enforce the local symmetry condition, we optimize the basis of vectors to minimize the atlas equivariance error of the map $\Phi$ on local charts.

Specifically, we first create a trainable tensor $B$, consisting of $k$ randomly initialized matrices of shape $m \times m$ where $m = \dim \mathcal{M}$. $B$ represents the Lie algebra basis to be discovered. Each basis vector is parametrized by a $m \times m$ matrix because, after exponentiation, it linearly transforms the flattened local neighborhoods of $\mathcal{M}$. In the training loop, we randomly sample an element $x$ from a dataset as well as a coefficient vector $\eta \sim \mathcal{N}_k(0, I)$. Using the coefficient vector and $B$, we have a group element: $g = \exp(\sum_{i=1}^k \eta_i B_i)$. The loss is the sum of $\mathcal{L}(\Phi_c(g \cdot x), g \cdot \Phi_c(x))$ over all $\Phi_c$, which measures the equivariance of each $\Phi_c$ with respect to the group element $g$. In this case, $\mathcal{L}$ is an error function appropriate to the context.

One problem with the given loss is that it often results in duplicate generators. Although cosine similarity is an establish regularization technique to avoid this issue (Yang et al., 2023; Forestano et al., 2023), it is sensitive to initial conditions and fails to produce consistent generators on consecutive runs. Therefore, we introduce the standard basis regularization instead, where one applies element-wise absolute value to each generator before applying the cosine similarity function. This incentivizes different vectors to share as little non-zero positions as possible, thereby driving the basis into standard form. We observe more interpretable results that are consistent across runs, albeit with a higher rate of duplicate generators. The standard basis regularization is provided below, where $|B|$ denotes element-wise absolute value and $\gamma$ is a positive weighting constant:

$$\mathcal{L}_{\text{sbr}}(B) = \gamma \sum_{i=1}^k \sum_{j=i+1}^k \frac{\text{vec}(|B_i|) \cdot \text{vec}(|B_j|)}{\|\text{vec}(B_i)\| \|\text{vec}(B_j)\|} \tag{3}$$

We prove a result about the global minima of $\mathcal{L}_{\text{sbr}}$ under certain conditions in Appendix A. We also list additional regularizations and a method for selecting the hyperparameter $k$ in Appendix B.

### 4.2.2 DISCOVERING DISCRETE SYMMETRIES

Many symmetry discovery methods only discover a Lie algebra basis, limiting the results to connected Lie groups. In practice, groups such as O(2) and the Lorentz group have multiple connected components, which is a natural consequence of discrete symmetries such as reflections. In this subsection, we introduce a method to discover discrete symmetries by identifying the $G_0$-cosets in the component group $G/G_0$.

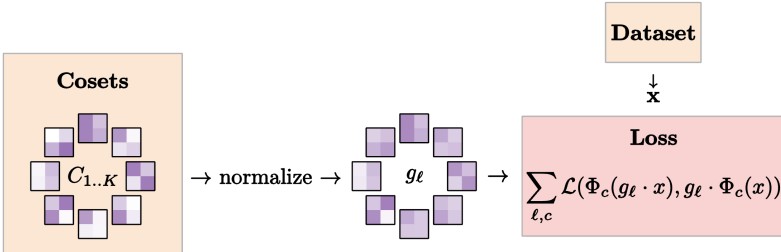

Figure 3: Discrete discovery training loop of ATLAS. All $K$ purple and white squares depict a matrix, and each matrix represents a discovered coset. The matrices are optimized under the given loss so that their normalized forms become elements of the ground truth cosets.

The discovery of these cosets faces several challenges. For one, we cannot parameterize the search space through a Lie algebra since there is no real-valued matrix that maps to an orientation reversing matrix via the exponential map. A discovery method operating on a Lie algebra is unable to realize both connected components of $O(2)$ since it includes a reflection. Moreover, even when the search space is set to $GL(n)$, we observe an abundance of local minima. Unless the seeded matrix is already close to a coset, it may fail to converge to anything useful.

To narrow the search space, we first assume the target group contains a finite number of connected components, which applies to most finite-dimensional Lie groups of interest. This implies we only need to consider transformations whose determinant has absolute value 1.

In the discovery process, we create a trainable tensor $C$ that contains representatives of $G_0$-cosets. $C$ is initially set to $K$ random matrices in $\mathbb{R}^{m \times m}$, where $K$ is chosen to be significantly larger than the expected number of cosets. Each $C_\ell$ is then independently optimized according to the loss of $\mathcal{L}(\Phi_c(\text{normalize}(C_\ell) \cdot x), \text{normalize}(C_\ell) \cdot \Phi_c(x))$ across all localized functions $\Phi_c$ and all $x \in \mathcal{X}$. The normalize function scales a matrix so that the absolute value of its determinant is 1.

After convergence, the top $q$ matrices in $C$ by loss value are taken to be the representatives of the ground truth cosets. We avoid duplicate cosets by comparing $C_i$ to $C_j$ and checking if $C_i C_j^{-1}$ belongs to the identity component, specified by the already discovered Lie algebra. In particular, we see if $\min_{t \in \mathbb{R}^k} \|C_i C_j^{-1} - \exp(\sum t_s B_s)\|_2 < \epsilon$ for a threshold $\epsilon$. After applying the filtration process, the final list comprises unique representatives of each coset of the target Lie group.

## 4.3 CONNECTION TO GAUGE EQUIVARIANT CNN

A related notion of local symmetry is introduced by Cohen et al. (2019) when defining gauge equivariant CNNs. In short, gauge equivariance implies that one should be able to arbitrarily orient the local coordinate systems used to define features and compute convolutions. Hence, it is a property of the *network* modeling the task function, rather than a property of the task function itself. This is a notable difference compared to our work, where the atlas equivariance group is intrinsic to the system and something that can be discovered.

The following theorem provides a concrete connection between gauge equivariance and atlas equivariance (proof in Appendix A).

**Theorem 1.** *Let $M$ be a gauge equivariant CNN that (a) has a linear gauge group $G$, (b) is $\mathcal{A}$ atlas local for some atlas $\mathcal{A}$ with trivial charts, and (c) operates on Euclidean space. Then, $M$ is $\mathcal{A}$ atlas equivariant to $G$.*

In practice, a gauge equivariant CNN is neither meant to operate on Euclidean space nor completely $\mathcal{A}$ atlas local. However, the result is approximately true for an arbitrary manifold as manifolds are locally flat. This implies that if a system is atlas equivariant for some group $G$, it is logical to set the gauge group of a downstream gauge equivariant CNN to $G$. We employ this technique as an application of our discovered symmetries below.

### 4.4 IMPLEMENTATION NOTES

Due to issues such as discretization, noise, boundary conditions, or limited a priori knowledge of a perfect atlas, real-world datasets may only be approximately, not exactly, $\mathcal{A}$ atlas local. To mitigate this issue, we sometimes allow the $\Phi_c$ predictors to look slightly outside of the associated chart, i.e. the radius of the input chart is higher than the radius of the output chart for any given $\Phi_c$. This provides the localized functions with additional context that may be missing from the unmodified input. Additionally, to avoid boundaries and awkward topologies (e.g. poles of a spherical mesh), we partially deviate from the definition of an atlas and do not require that the charts fully cover the manifold. Empirically, if the charts span the majority of $\mathcal{M}$ rather than fully covering $\mathcal{M}$, our method is still able to discover local symmetries within the given region.

## 5 EXPERIMENTS

We experiment on a few tasks to validate our methodology and implementation. Specifically, we perform experiments on (1) top-quark tagging task for direct comparison with global symmetry discovery baselines; (2) synthetic partial differential equation to test our model's sensitivity to various atlases; (3) projected MNIST classification and ClimateNet weather segmentation tasks to highlight our success in the discovery of atlas equivariances as well as the performance gains when discovered symmetries are incorporated into downstream models. Additional details about each experiment, such as chart sizes and other hyperparameters, are present in Appendix C.

### 5.1 GLOBAL SYMMETRY COMPARISON

To directly compare our method to existing discovery pipelines that focus on global symmetries, we first attempt to learn global invariances in the top quark tagging experiment. Specifically, we compare our results with those of LieGAN (Yang et al., 2023).

The goal is to classify between top quark and lighter quarks jets present in the Top Quark Tagging Reference Dataset (Kasieczka et al., 2019). The dataset contains 2M observations, consisting of four-momentum of up to 200 particle jets. The classification is invariant to the entire Lorentz group $O(1,3)$, which we will try to discover. We use our infinitesimal generator discovery pipeline to learn the invariances of the predictor. We seed our basis with 7 generators. In Figure 4, we show that the discovered basis matches closely with that of $\mathrm{SO}^+(1,3)$, the identity component of the Lorentz group. Moreover, computing the invariant tensor using the method from Yang et al. (2023), we find that the invariant tensor has a cosine correlation of $0.9996$ with the ground truth Minkowski tensor $\mathrm{diag}(-1,1,1,1)$. This is a strong result, slightly superior to LieGAN's cosine correlation of $0.9975$.

We then try to discover the various cosets of the symmetry group. We seed our discovery process with $K = 256$ matrices. In the dataset, the time component of all momenta are positive, and hence it

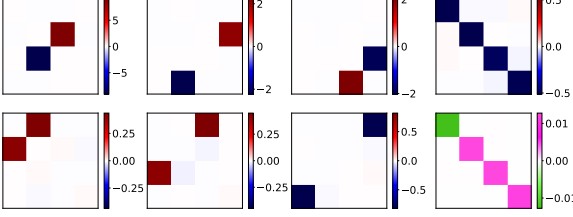 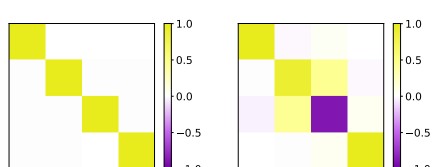

Figure 4: ATLAS discovers infinitesimal generators for $\mathrm{SO}^+(1,3)$ in the top tagging task. Here, each red and blue heatmap denotes a Lie algebra basis element. For each generator, the value of its entries are depicted by the individual colors. Generators 0, 1, 2 correspond to $\mathrm{SO}(3)$ rotation and generators 4, 5, 6 indicate boosts. Generator 3 indicates uniform scaling, which is not part of $\mathrm{SO}^+(1,3)$. The bottom right heatmap displays the computed invariant metric.

Figure 5: ATLAS discovers two cosets in the top tagging experiment: a representative from the identity and parity component. Each yellow and purple heatmap depicts a coset's representative in matrix form, where the colors denote the values of that matrix's entries.

is difficult to find the time-reversal generator of the Lorentz group. However, in Figure 5, we discover a parity transformation, which means that the entire learned symmetry group is $O^+(1, 3)$. In contrast, LieGAN is only able to discover connected Lie groups and hence only reports the identity component $SO^+(1, 3)$.

Finally, we use the computed invariant metric tensor as an inductive bias to construct a well-performing classification model. Specifically, we create AtlasGNN by modifying LorentzNet (Gong et al., 2022) to use our discovered metric instead of the Minkowski tensor. In Table 1, we observe better accuracy and AUROC than many baselines and nearly match LorentzNet, which uses ground truth symmetry.

Table 1: Downstream test results for top tagging task. Baselines results are from Yang et al. (2023); Gong et al. (2022).

| MODEL | ACCURACY | AUROC |
|---|---|---|
| LORENTZNET | 0.941 ±0.0010 | 0.9862 ±0.0004 |
| **ATLASGNN** | 0.939 ±0.0002 | 0.9852 ±0.0001 |
| LIEGNN | 0.938 ±0.0001 | 0.9849 ±0.0001 |
| LORENTZNET (W/O) | 0.935 ±0.001 | 0.9835 ±0.0003 |
| EGNN | 0.925 ±0.0001 | 0.9799 ±0.0004 |

## 5.2 PARTIAL DIFFERENTIAL EQUATION

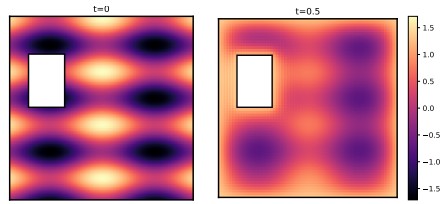 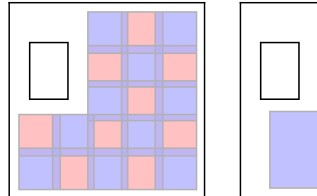 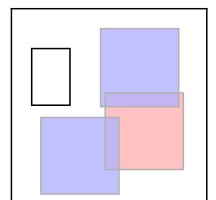

(a) Example input and output field for heat experiment.

(b) Depiction of atlases for heat experiment. Each blue or red square represents an individual chart.

Figure 6: Illustration of PDE experiment settings.

Next, we want to see if our method can indeed learn atlas equivariances and also measure its sensitivity to various atlas configurations. Specifically, we experiment if our model can discover the local symmetries of the heat equation $\frac{\partial u}{\partial t} = \alpha(\frac{\partial^2 u}{\partial x^2} + \frac{\partial^2 u}{\partial y^2})$ in $\mathbb{R}^2$ (Figure 6a). The task function in this case simulates heat flow for $0.5$ seconds given an initial condition. In the simulation, we exclude a certain rectangular region and treat it as a heat source, thereby breaking any global symmetry. However, sufficiently far from any boundary, the system still exhibits $O(2)$ local symmetry.

To test the sensitivity of our method to different atlases, we perform our experiments with one atlas containing 19 charts and another containing 3 charts (Figure 6b). In either case, we seed the model with a single infinitesimal generator and $K = 16$ cosets and report the unique cosets from the top $q = 8$. In Figure 7, we demonstrate that ATLAS is able to accurately recover the $O(2)$ atlas equivariance group in both situations. However, the first atlas does slightly outperform the second.

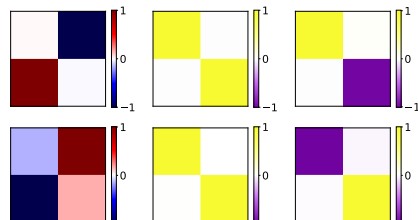

Figure 7: PDE discovered results. The top row depicts the results using the first atlas, whereas the results using the second atlas are given below. In each case, the leftmost entry shows the discovered infinitesimal generator, and the right two columns are the discovered cosets. Both atlases lead to an approximate generator for $SO(2)$ as well as a coset containing a reflection.

## 5.3 MNIST ON SPHERE

To highlight the benefits of using our learned results in downstream models, we design a projected MNIST segmentation task. In this experiment, we project three digits from the MNIST dataset onto

a sphere (Figure 8). Before projection, each image is randomly rotated up to 60 degrees clockwise or counterclockwise. The goal of the model is to classify each pixel as either the background or its numeric value. Although the rotation of the digits adds a local symmetry, there is no continuous global symmetry since the position of each of the three digits is fixed. A natural atlas to use in this problem is assigning a single chart to the region of each digit. We then train a predictor for each of the three charts using CNNs. In the discovery process, we seed our model with a single infinitesimal generator. To demonstrate the benefit of considering local symmetry, we compare our results against a modified LieGAN that represents global symmetries as subgroups of SO(3).

After running the discovery process, we find an approximate $SO(2)$ generator: $\begin{bmatrix} -0.03 & -1.00 \\ 1.00 & 0.02 \end{bmatrix}$. In Figure 9, we show that applying a local transformation suggested by ATLAS leads to a non-trivial change, but one that still preserves the form of the dataset. On the other hand, the global transformation sampled from LieGAN's result clearly modifies the input out of distribution, suggesting the result is a random rotation rather than an actual symmetry. This highlights a case where considering local symmetry is more appropriate than searching for global symmetry.

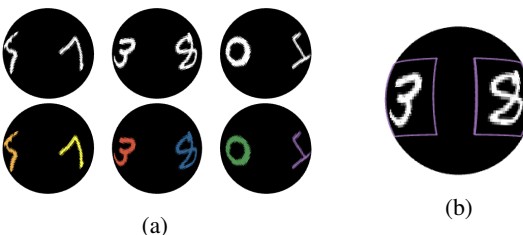

(a)                                         (b)

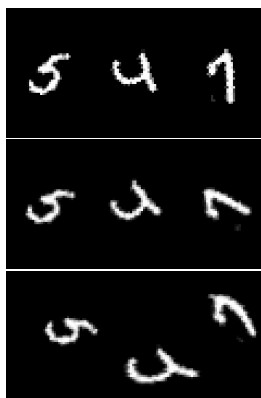

Figure 8: MNIST experiment setup. (a) The input feature fields (top) are given by three digits rotated then projected onto the equator of a sphere. To construct the output feature fields (bottom), the model must label each pixel as either background or its numeric value if it is a part of a number. (b) We highlight two of the charts used in our atlas.

Figure 9: Local and global transformations on MNIST. In the upper row, we highlight an element from the dataset in its projected form. In the middle row, we apply a local transformation based upon ATLAS's discovery. The last row is the result after applying a global rotation suggested by LieGAN.

Table 2: Test Accuracy of MNIST downstream segmentation task. We exclude pixels that are labeled background.

| MODEL | ACCURACY |
|---|---|
| REGULAR CNN | 0.6975 |
| SO(2) GAUGE CNN | 0.9381 |

In addition, we construct a gauge equivariant CNN using the discovered $SO(2)$ group and compare it to a regular CNN. In an arbitrary manifold, it may be difficult to perform strided convolutions or pooling. Therefore, we subject both models to this constraint as well. We train each model on a dataset where the digits are rotated $\pm 60$ degrees, and test it on one where digits are rotated $\pm 180$. As seen in Table 2, the gauge equivariant network clearly outperforms the vanilla CNN, as the inductive bias allows it to generalize outside of its training set.

## 5.4 CLIMATENET

For our final experiment, we evaluate our method on a real-world dataset, ClimateNet, proposed by Prabhat et al. (2021). Each input in the dataset contains 16 atmospheric variables across the surface of the Earth, and the output is a human label to determine whether each pixel is part of the background, an atmospheric river, or a tropical cyclone. We aim to discover the atlas equivariance group.

We use an atlas that has 4 charts spread through the surface of the earth. When we seeded our model with 1, 2, or 3 infinitesimal generators, we find that the resultant basis is not similar across consecutive runs. This suggests that the symmetry group is actually 4-dimensional. To confirm this, we plot a chart predictor's output after applying various linear actions in Figure 11. The figure highlights

that the predictor is mildly equivariant to a wide range of actions. In fact, Figure 10 demonstrates comparable magnitudes of the 4 generators, suggesting that all directions are equally strong in terms of symmetry. All of these points to evidence that the atlas equivariance group is $GL^+(2)$.

We compare using the discovered atlas equivariance group to the structure group in a downstream gauge equivariant CNN. Specifically, we use an $r = 6$ icoCNN architecture in two different settings (Diaz-Guerra et al., 2023). For the baseline, we set the gauge group of the icoCNN to be $SO(2)$ (the structure group). While it is not easy to construct a gauge equivariant CNN using steerable kernels (Weiler & Cesa, 2019) for a non-compact group such as $GL^+(2)$, the closest approximation is to have the kernel be spatially uniform. That is, all values for a given input-output channel pair are the same for a particular filter. In Table 3, we show that the "flat" kernel CNN is able to match the baseline performance despite having 7 times fewer parameters. This highlights a benefit to using the discovered group as the gauge group versus choosing the structure group of the manifold.

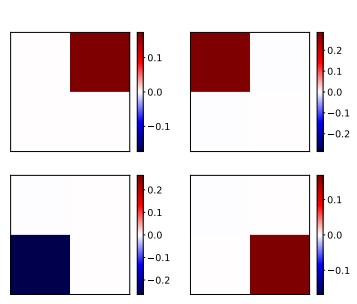

Figure 10: ATLAS discovers $GL^+(2)$ symmetry in ClimateNet dataset.

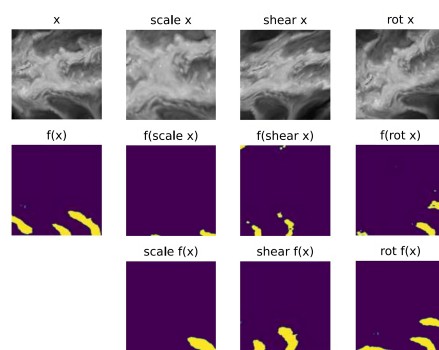

Figure 11: The inputs and outputs of a ClimateNet predictor after applying various transformations. The plotted $x$ values are visualizations of TMQ. In the outputs, purple indicates background and yellow represents atmospheric river.

Table 3: ClimateNet dataset accuracy results. We compute the IoU (intersection over union) obtained for the baseline $SO(2)$ gauge group model (1st row), our $GL^+(2)$ gauge group model (2nd row), and between human experts (3rd row). We include the last row to demonstrate that the human labelers have a degree of disagreement, providing context for low IoU scores. See Appendix C for full details.

| MODEL | PARAMS ↓ | BG ↑ | TC ↑ | AR ↑ | MEAN ↑ | PRECISION ↑ | RECALL ↑ |
|---|---|---|---|---|---|---|---|
| $SO(2)$ MODEL | 766K | 0.9107 | 0.1744 | 0.3839 | 0.4896 | 0.5983 | 0.6274 |
| $GL^+(2)$ MODEL | 111K | 0.9086 | 0.1720 | 0.3790 | 0.4865 | 0.5846 | 0.6344 |
| HUMAN | - | 0.9137 | 0.2475 | 0.3467 | 0.5026 | - | - |

## 6 CONCLUSION

In this paper, we introduce atlas equivariance and propose **au**tomatic **l**ocal **s**ymmetry discovery (ATLAS) as an architecture capable of learning local symmetries for a variety of systems. We demonstrate that our methodology can discover both infinitesimal generators and cosets of the atlas equivariance group from a dataset. Moreover, the results show that the atlas equivariance group also serves as an inductive bias in downstream gauge equivariant networks. This proves the need to focus on local symmetries of a system as opposed to solely their global ones.

While our method effectively discovers atlas equivariances, we should note that atlas equivariances only describe a subset of all possible symmetries of a manifold. We set the stage for future work to explore the discovery of larger groups. In addition, while we show that ATLAS is resilient to slight modifications of the given atlas, a priori knowledge of a suitable atlas is still important and may not always be available. An extension to our work can relax this condition by developing a method that discovers the atlas in tandem to the atlas equivariance group. Another possible direction for future work is to consider the symmetries that act on both the manifold and the features, such as the point symmetries in partial differential equation systems.

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

# A  PROOFS

## A.1  ATLAS EQUIVARIANCE OF GAUGE EQUIVARIANT CNN

**Theorem 1.** *Let $M$ be a gauge equivariant CNN that (a) has a linear gauge group $G$, (b) is $\mathcal{A}$ atlas local for some atlas $\mathcal{A}$ with trivial charts, and (c) operates on Euclidean space. Then, $M$ is $\mathcal{A}$ atlas equivariant to $G$.*

*Proof.* We first clarify a few definitions. By trivial charts, we simply mean that $\varphi_c$ is the inclusion map. We model a gauge equivariant CNN as a series of convolutional layers and pointwise nonlinearities (Cohen et al., 2019). To enforce gauge equivariance, we will require all kernels $K : \mathbb{R}^d \to \mathbb{R}^{C_{\text{in}} \times C_{\text{out}}}$ in the network to satisfy $K(g^{-1}v) = K(v)$ for $g \in G$.

To show that such a network $M$ is $G$ atlas equivariant, we must prove that there exists $G$-equivariant localized functions $M_c$. Recall that in the definition of $\mathcal{A}$ atlas local, $M_c$ has no restriction on its output field outside $\varphi_c(U_c)$. Consequently, since all charts are trivial and $M$ is assumed to be $\mathcal{A}$ atlas local to begin with, $M$ itself is suitable for each $M_c$.

It remains to show that $M$ is $G$ equivariant. Indeed, a $G$ gauge equivariant convolutional layer on Euclidean space as presented above is equivalent to a $G$-steerable convolution (Weiler & Cesa, 2019). Moreover, $G$-steerable convolutions are globally equivariant. As we assume feature vectors transform trivially in response to a group action, all pointwise nonlinearities are automatically $G$ equivariant. $M$, the composition of $G$ equivariant layers, is then $G$ equivariant.

Thus, $M$ is $\mathcal{A}$ atlas equivariant with respect to $G$. $\qquad\square$

## A.2  ARGMIN OF STANDARD BASIS REGULARIZATION

Let $V$ be a $k$-dimensional subspace of $\mathbb{R}^n$. We call a basis $\{b_i\}_{i=1}^k$ of $V$ *disjoint* if the set of indices of all non-zero elements of $b_i$ and the similar set for $b_j$ are disjoint whenever $i \neq j$. The following theorem gives a result about the arguments of the minima of $\mathcal{L}_{\text{sbr}}$.

**Theorem 2.** *Let $V$ be a $k$-dimensional subspace of $\mathbb{R}^n$ for which there exists a disjoint basis. Then, among all possible bases $\{b_i\}_{i=1}^k$ of $V$, $\mathcal{L}_{sbr}(b)$ is minimal if and only if $b$ is disjoint.*

*Proof.* Suppose $\{b_i\}$ is a basis of $V$. If $b$ is disjoint then for all $i \neq j$ we have $|b_i| \cdot |b_j| = 0$ since for each index $\ell$, at least one of $b_{i,\ell}$ or $b_{j,\ell}$ will be zero. Conversely, if $b$ is not disjoint, then there exist some $i \neq j$ such that $|b_i| \cdot |b_j| > 0$. To see this, note that we must have some $i \neq j$ where $b_i$ and $b_j$ share a non-zero element at index $\ell$. $|b_i| \cdot |b_j|$, the sum of non-negative numbers, is then greater or equal to $|b_{i,\ell}||b_{j,\ell}| > 0$.

In particular, this implies that for a disjoint basis $b$, we have $\mathcal{L}_{\text{sbr}}(b) = 0$, but otherwise $\mathcal{L}_{\text{sbr}}(b) > 0$. By assumption, there exists at least one disjoint basis so then the minimum of $\mathcal{L}_{\text{sbr}}$ over all possible bases of $V$ is 0. This minimum is attained exactly when the input basis is disjoint. $\qquad\square$

# B  ADDITIONAL IMPLEMENTATION DETAILS

A common degenerate solution in discovering the Lie algebra basis is when all basis vectors tend towards 0, corresponding to the identity transformation. To prevent this, we add the following growth regularization, where $\iota$ and $\beta$ are hyperparameters.

$$\mathcal{L}_{\text{gr}} = -\iota \sum_{i=1}^{k} \min(\|B_i\|, \beta)$$

The min term ensures that the model does not produce arbitrarily large generators. In the experiment details, we refer to $\iota$ as the growth factor and $\beta$ as the growth limit.

An important hyperparameter in the discovery of infinitesimal generators is $k$, the dimension of the basis. Forestano et al. (2023) suggest setting the dimension as the highest number that results in a

vanishing loss. However, we find that the threshold for what constitutes "vanishing" can become ambiguous in real-world datasets. Therefore, to determine the final value of $k$, we first run the model repeatedly, varying the basis dimension in different runs. We initially set $k$ as the minimum value such that a model with $k$ generators always converges to the same algebra, irrespective of the starting conditions. Then, we increment $k$ one-by-one until the norm of the weakest generator drops below a threshold.

# C  EXPERIMENT DETAILS

In this section, we include some additional details for the performed experiments, including the hyperparameters and synthetic dataset configurations.

## C.1  GLOBAL SYMMETRY DISCOVERY

The predictor for this task is a 3-layer MLP that takes the input of 30-leading constituents for each sample, constructed of 4-momenta ($E/c, p_x, p_y, p_z$). This results in an input dimension of 120. The predictor is trained for 10 epochs with a learning rate of 0.001 prior to the discovery process. We use cross-entropy loss for training. We find that the predictor can be relatively naive and still be suitable for symmetry discovery.

In the infinitesimal generator discovery, we seed the basis with 7 elements by our criteria for choosing the dimension of the basis. Although the Lorentz group is 6 dimensional, our model occasionally finds an additional scaling generator. Interestingly, Yang et al. (2023) find a similar generator using their methodology. We run the model for 10 epochs using cross-entropy loss. We set the coefficient of standard basis regularization to be 0.1 and the growth factor of the generators to be 1. We do note set a growth limit. The learning rate is 0.001.

For coset discovery, we seed the model with $K = 256$ basis elements and run 3 epochs with cross-entropy loss. We believe $K$ can be reduced significantly and still discover the parity component. However, apart from computational time, we do not observe any downsides to keeping the number high. To filter out the final representatives, we find all the unique cosets in the top $q = 16$ matrices. The learning rate is 0.001.

In the downstream task, we replace all Minkowski norms and Minkowski inner products of LorentzNet Gong et al. (2022) with those appropriate to our discovered metric. We construct the model with 6 group equivariant blocks with 72 hidden dimensions and train it with a batch size of 32 for 35 epochs with dropout rate of 0.2, weight decay rate of 0.01, and learning rate of 0.0003. Note that while our predictor used for symmetry discovery was limited to the 30 leading components, the downstream model does not face the same restriction. We run our model and the baselines 3 times and record the average and standard deviation in Table 1.

## C.2  PARTIAL DIFFERENTIAL EQUATION

For this experiment, we create a dataset of 10000 samples, each of size 128x128. The exclusion region spans from $(0.1, 0.2)$ to $(0.3, 0.5)$, where $(0, 0)$ is the top left, and $(1, 1)$ is the bottom right. The initial condition is given by creating a purely vertical sinusoid with random parameters and adding it to a purely horizontal sinusoid with random parameters. To construct the output for each input, we approximate the heat equation using a finite difference method. We use $\alpha = 1$. In particular, we numerically integrate 50 times with $dt = 0.01$. We use the Dirichlet boundary condition, where all boundary values (including those on the excluded region) take the value $\sqrt{2}$.

The charts in the first atlas have an in-radius of 14 pixels (full dimension 29x29) and an out-radius of 10 (full dimension 21x21). There are a total of 19 charts centered at the following locations specified in the previously defined coordinate space: (0.5, 0.15), (0.675, 0.15), (0.85, 0.15), (0.5, 0.325), (0.675, 0.325), (0.85, 0.325), (0.5, 0.5), (0.675, 0.5), (0.85, 0.5), (0.15, 0.675), (0.325, 0.675), (0.5, 0.675), (0.675, 0.675), (0.85, 0.675), (0.15, 0.85), (0.325, 0.85), (0.5, 0.85), (0.675, 0.85), (0.85, 0.85). The $\varphi_c$ do not perform any distortion, but do recenter each chart.

The charts in the second atlas have in-radius 26 (full dimension 53x53) and out-radius 20 (full dimension 41x41). They are centered at the following locations: `(0.65, 0.3)`, `(0.675, 0.625)`, `(0.35, 0.75)`. The $\varphi_c$ act the same way as in the first chart.

The predictors are simple 4-layer CNNs. They are trained for 10 epochs in tandem with the discovery process. In the discovery process we seed the model with a single infinitesimal generator. The growth factor is set to 0.1 and growth limit is 1. We use mean absolute error as the loss. We seed the model with $K = 16$ cosets and took the top $q = 8$ matrices before filtering duplicates. We run our model for 10 epochs. The learning rate is 0.001.

## C.3 MNIST ON SPHERE

The dataset is constructed by creating 10000 spheres. Each sphere has 3 randomly selected digits from the MNIST dataset projected onto its equator at fixed positions. In particular, we first rotate each of the three digits $\pm 60$ degrees. Then, all three digits of size 28x28 are placed onto a cylinder of dimensions 120x60 at equal intervals. Finally, they are projected onto a sphere using an equirectangular projection. To compute the output sphere, we label all pixels that are fully black as background. The pixels that have non-zero color are labeled with their numeric value. Consequently, there are a total of 11 classes.

The chosen atlas uses 3 charts located at the locations of each of the three digits. In particular, the in- and out-radius of each chart is 14 (full dimension 29x29). The predictors for each chart are CNNs that are identical in architecture but independently trained. When training the predictors, we use cross-entropy loss and weigh background pixels 10 times less than numeric pixels. The growth factor of the generator is set to 0.35 and growth limit is 1. The predictors are trained in tandem with the discovery process. In particular, we run the discovery process for 20 epochs with a learning rate of 0.001. As a baseline, we compare to a modified LieGAN that can discover subgroups of $\mathrm{SO}(3)$. LieGAN is given a single continuous generator as well. LieGAN is run for 20 epochs with a learning rate of 0.0002 for the discriminator and 0.001 for the generator.

In the downstream task, we train two CNNs that are identical in design, except that one has $Z_4$ steerable kernels. The model that has $Z_4$ steerable kernels has less than a third of the parameters of the unmodified CNN. Both models are trained for 100 epochs on the dataset. During training, to compensate for the abundance of background pixels, we weigh the background pixels 0.005 times as much as the numeric counterparts. In reporting accuracy, we fully ignore the background pixels and focus only on the numeric pixels.

## C.4 CLIMATENET

We use ClimateNet dataset, which is an expert labeled open dataset provided by Prabhat et al. (2021). There are roughly 200 input images in the training set, with some images having multiple human expert labelers.

In the symmetry discovery process, we use an atlas of 4 partially overlapping charts that are scattered across the equator. The in-radius is set to 200 (full dimension 401x401) whereas the out-radius is 150 (full dimension 301x301). The individual $\varphi_c$ do not do any additional projection, i.e. they keep the projection that the dataset used to parameterize the sphere as a rectangle. We use a modified CGNet (Wu et al., 2018; Prabhat et al., 2021) as the predictor for each chart. In particular, it is given four atmospheric variables as input: TMQ, U850, V850, PSL. The predictors are trained in tandem with the discovery process. The model is seeded with 4 generators, and we use a batch size of 16 and run for 30 epochs using cross-entropy loss. The coefficient of the standard basis regularization is set to 0.05, the growth factor is 5.0, and the growth limit is 1.0. The learning rate is 0.001.

The downstream models are implemented using a U-net (Ronneberger et al., 2015) version of icoCNN (Diaz-Guerra et al., 2023). We also add strided convolutions and replace layer norm with batch norm. Since batch norm is typically not equivariant, we perform average pooling beforehand when necessary. We set the resolution of the icosahedron to $r = 6$. We train each model for 20 epochs with batch size of 4 with a learning rate of 0.001. Note that in the downstream models, we give them all 16 atmospheric variables.

We elaborate on the results of table 3. For the first two rows, the mean IoU, precision, and recall are calculated between the model predictions and every human expert label that exists for that image and then averaged. Then, these results themselves are averaged across all input images in the test dataset Prabhat et al. (2021). We run each model 10 times and include the run with the highest mean IoU. In the third row, we compute the mean IoU between human labels for the same input image in the training set. All scores are computed after projection onto an icosahedron.

# D  ADDITIONAL EXPERIMENTS

## D.1  COSET DISCOVERY ON A SYNTHETIC PROBLEM

We want to see if ATLAS can discover multiple cosets in situations where the loss of a one coset (typically the identity) is lower than the loss of another coset in the symmetry group. This could happen when the trained predictor is not super accurate or if the symmetry of the true function itself is not perfect. To do so, we experiment on the global discrete symmetries of the function $f(x, y) = \arctan\left(\frac{y+0.1}{x}\right)$. The identity transformation is a perfect symmetry of $f$ and rotation by 180 degrees is an approximate symmetry. Although there is some variability between runs, Figure 12 highlights we are able to consistently discover rotation coset representatives among the top 24 cosets.

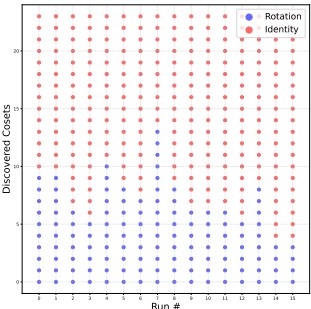

Figure 12: Discovered cosets of arctan symmetry group. In each column, we plot the distribution of the top 24 cosets for a given run. Red denotes the identity component and blue is the rotation component.

## D.2  TOP-TAGGING EXPERIMENT WITHOUT STANDARD BASIS REGULARIZATION

To test how effective $\mathcal{L}_{\text{sbr}}$ is at regularizing a basis towards standard form, we retry the the infinitesimal generator discovery of the top-tagging experiment. This time, we replace $\mathcal{L}_{\text{sbr}}$ with cosine similarity and plot the resultant basis in Figure 13.

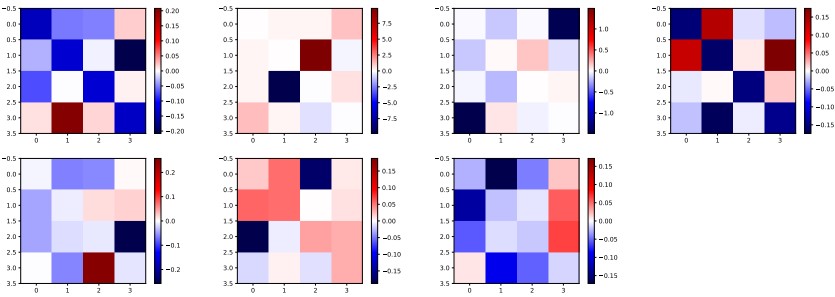

Figure 13: Top-tagging basis using cosine similarity.

In this basis, all 21 pairs of generators share at least one non-zero term. On the other hand, the one produced by $\mathcal{L}_{\text{sbr}}$ (Figure 4) has a single such pair. We conclude that $\mathcal{L}_{\text{sbr}}$ is effective at forming a standard basis.

### D.3 TOP-TAGGING AND PDE EXPERIMENT WITHOUT COSET NORMALIZATION

We want to see the usefulness of the normalization step during coset discovery. To do so, we repeat the discrete discovery step of both the top-tagging and PDE experiments without normalization.

In the top-tagging experiments, we discover two cosets (Figure 14). There is more noise and a slight scaling factor as compared to the result with normalization (Figure 5), but we are still able to discover the parity and identity components. In the PDE experiment, we are usually able to discover the identity and reflection components (Figure 15), but the filtration process fails and extraneous cosets are also included.

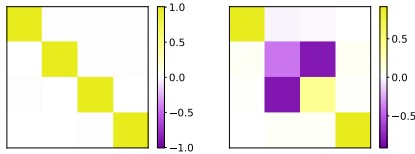

Figure 14: Top-tagging cosets without normalization.

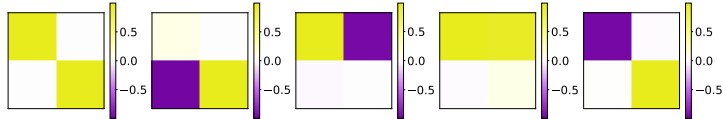

Figure 15: PDE cosets without normalization.

In both cases, we manage to find the ground truth cosets but there is more noise overall. We conclude that normalization is a helpful step of the discrete discovery process.

### D.4 COMPARISON WITH LIEGG IN PDE EXPERIMENT

To further highlight the necessity of considering local symmetry, we compare our results from the PDE experiment to those of LieGG (Moskalev et al., 2022). We generalize LieGG to be able to discover global equivariances when the group acts on $\mathbb{R}^2$. Specifically, we treat our dataset as modelling a collection of input and output feature fields $(F_\ell, G_\ell)$ where $F_\ell, G_\ell : \mathbb{R}^2 \to \mathbb{R}$. This is given to us in discretized form so that we only know $F_\ell$ and $G_\ell$ along some sampling $(x_1^1, x_1^2), (x_2^1, x_2^2) \ldots (x_n^1, x_n^2)$ of $\mathbb{R}^2$. To construct the polarization matrix, for every single output location $x_i$ in every single datapoint $(F_\ell, G_\ell)$, we add a row given by the following linear equation (4). The equation is modified from the original equation (3) from Moskalev et al. (2022) to learn *equivariance* instead of *invariance*. We keep the notations consistent with those in Moskalev et al. (2022). We note that this procedure is rather space-intensive as the number of rows is proportional to the number of total output pixels.

$$\sum_{j,k \in \{1,2\}} \mathfrak{h}_{k,j} \left[ x_i^j \frac{\partial G_\ell(x_i)}{\partial x_i^k} - \sum_{p=1}^n x_p^j \frac{\partial F_\ell(x_p)}{\partial x_p^k} \frac{\partial G_\ell(x_i)}{\partial F_\ell(x_p)} \right] = 0 \tag{4}$$

For simplicity, we consider the setting where LieGG is given access to the ground truth partial differential equation instead of a predictor network. We also do not perform time stepping and focus solely on the global symmetries of $\frac{\partial u}{\partial t} = \alpha(\frac{\partial^2 u}{\partial x^2} + \frac{\partial^2 u}{\partial y^2})$. This setup, while slightly different from the experiments for our method, only makes it easier for LieGG to learn the symmetry. The singular values of the discovered generators are shown in Figure 16 .

When we remove the heat source and there is a true global symmetry, the smallest singular value is $1.137 \cdot 10^{-6}$ and is associated with the $SO(2)$ generator. When there is a heat source like the one in the PDE experiment, all singular values become much higher. We conclude that global symmetry discovery methods such as LieGG are unable to discover meaningful symmetries in systems that only have local symmetries.

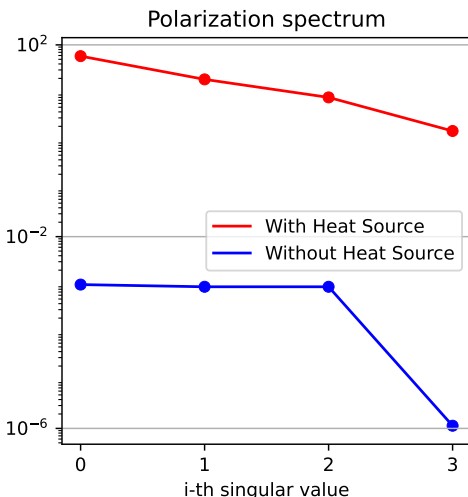

Figure 16: Singular values of generators discovered by LieGG in modifed PDE experiment. We run twice: once without the rectangular heat source (blue) and once with the heat source (red).

# E    ALGORITHM ANALYSIS

We analyze the space and time complexity of the different parts of our algorithm. In our analysis, we assume that $\dim \mathcal{M}$ is fixed as a constant. We use the notation that $T$ is the number of training iterations for a given run, $k$ is the dimension of the ground truth symmetry, $P$ is the number of localized predictors, $K$ is the total number of cosets used during training, and $q$ is the max number of cosets reported.

## E.1    INFINITESIMAL GENERATORS

Our algorithm requires storing the discovered Lie algebra basis, which takes space proportional to $k$. We also need to store each of the $P$ localized predictors, giving us space complexity of $\mathcal{O}(k + P)$.

We first consider the amount of time a single training step takes. Calculating the standard basis regularization takes $\mathcal{O}(k^2)$ time. For a given predictor, computing the main loss takes time proportional to $k$, which is needed for the sampling of the group element. In our implementation, we evaluate all $P$ predictors in a training step. There are $T$ total training steps in a given run. Then, we require at most $k$ total runs to determine the optimal dimension of the basis. This gives the total time complexity as $\mathcal{O}(kT(k^2 + kP)) = \mathcal{O}(k^2T(k + P))$

The above result is somewhat misleading as it hides the high constant factor that the predictor evaluation entails. If we ignore the regularizations and only consider the predictor evaluation, the time complexity becomes $\mathcal{O}(kTP)$.

## E.2    DISCRETE SYMMETRIES

We require storing the $K$ cosets as well as the $P$ predictors. The space complexity is then $\mathcal{O}(K + P)$.

In each training step, we evaluate each of the $P$ predictors on $K$ transformed inputs, corresponding to the $K$ cosets. This is repeated for all $T$ training iterations. After the training process, we must filter the duplicate cosets. In the worst case, we report $q$ cosets in which case we need to do $\mathcal{O}(Kq)$ comparisons. The total time is then $\mathcal{O}(K(TP + q))$

