# OpenReview forum: "ATLAS: Automatic Local Symmetry Discovery"
_ICLR.cc/2025/Conference — Submitted to ICLR 2025_

### Official Review · Reviewer_51D4 · 2024-10-29

**Soundness:** 3
**Presentation:** 3
**Contribution:** 2
**Rating:** 6
**Confidence:** 3

**Summary:**

Formalizing the notion of local symmetry as atlas equivariance, the authors proposed an automatic local symmetry discovery approach (ATLAS) to recover the local symmetries of a function using local predictor networks. They learned a Lie group basis to ensure that the predictors are equivariant, resulting in the discovery of local symmetry groups with multiple connected components. This approach was tested in experiments on top-quark tagging, synthetic partial differential equations, classification, and climate segmentation.

**Strengths:**

**S1.** The paper is very well-written and presented, with a well-defined motivation for studying local symmetries.

**S2.** The development of a method to discover atlas equivariance in a given data distribution in the form of a Lie group is unique.

**S3.** The experiments covered diverse datasets, including tasks like classification and segmentation, demonstrating the model's applicability across various domains.

**Weaknesses:**

**W1.** The assumption of limiting the target group to a finite number of connected components requires further justification and additional experimentation, if possible. Lie groups can have an infinite number of components in some of the extensively researched areas like diffeomorphisms. I assume that this approach does not hold for such cases, which could limit the generalizability of the method to certain pre-defined knowledge. I would encourage the authors to shed more light on this limitation and to experimentally validate why the model might fail to converge, as they have suggested.

**W2.** As the downstream test results show very minor differences between the baselines and the proposed approach, what are the advantages of using this approach? Also, I would suggest the authors to experiment with some real-world image datasets.

**W3.** While the authors extensively focused on the inability of existing SOTA models to capture local neighborhoods, it would be great if the authors could experimentally compare them and validate this, specifically with models such as (Moskalev et al., 2022), (Dehmamy et al., 2021), LieGAN, and similar works that capture global symmetries. Comparing these models would effectively validate the authors' assumptions.

**W4.** I believe the incorporation of the symmetries discovered by ATLAS leading to parameter efficiency has not been validated. I request the authors to discuss how they ensure effective parameter efficiency with their proposed approach in the rebuttal.

**W5.** How can we randomly sample a localized function $\phi_c$ from $\phi$? Do we need to consider the random bias that might be introduced through the random sampling of $\phi_c$? Besides, is there any potential adversarial impact of the randomly sampled coefficient vector $\eta$ in terms of deriving the group element $g$?

**Minor**
- How the parameter $\epsilon$ is being selected? An ablation would be great to analyze the robustness of this parameter over the model's performance.


While there are some experimental concerns and methodological confusion, the current rating reflects the accumulated efforts in problem definition, motivation, presentation, and diverse experimentation. I strongly suggest that the authors consider all of the findings in their rebuttal. Good luck.

**Questions:**

I tried to cover most of my concerns and questions in the *Weaknesses* section. I kindly request the authors to review that section.

---

> ### Author Response · Authors · 2024-11-20
> **Response to Weakness 1-4**
>
> > The assumption of limiting the target group to a finite number of connected components requires further justification and additional experimentation, if possible.
>
> In general, many of the most common symmetry groups (including $\mathrm{O}(N)$, $\mathrm{SO}(1,3)$, and $\mathrm{GL}(N)$) have a finite component group. It is certainly true that Lie groups may have an infinite number of cosets, but we believe this describes the symmetries for only a small fraction of real-world systems.
>
> Regardless, when we retry the top-tagging experiment without normalization (thereby including Lie groups with an infinite number of connected components in the search space), we are still able to discover the parity and identity components, albeit with higher noise. When we do the same for the heat experiment, we correctly find representatives of the identity and reflection cosets, but the filtration process occasionally fails and reports more than $2$ total cosets. Thus, the normalization step is helpful, but not critical, to our method. If one wants to consider Lie groups with infinite components, we believe it would only require minor tuning for ATLAS to be able to discover them.
>
> > As the downstream test results show very minor differences between the baselines and the proposed approach, what are the advantages of using this approach? Also, I would suggest the authors to experiment with some real-world image datasets.
>
> In terms of downstream models, the primary advantage of our approach is that one can construct a gauge equivariant network without knowing the symmetry group a priori. We validate this in experiment 3, where the downstream network using our discovered result achieves 24% higher accuracy in comparison to a non-equivariant model. In experiment 4, the downstream model demonstrates increased parameter efficiency over the baseline.
>
> We are interested in the suggestion to explore real-world image datasets. Do you have a particular dataset in mind that you believe demonstrates local symmetry?
>
> > While the authors extensively focused on the inability of existing SOTA models to capture local neighborhoods, it would be great if the authors could experimentally compare them and validate this, specifically with models such as (Moskalev et al., 2022), (Dehmamy et al., 2021), LieGAN, and similar works that capture global symmetries. Comparing these models would effectively validate the authors' assumptions.
>
> We compare our discovered results against LieGAN in the third experiment. We also hope to follow-up with comparisons against LieGG or L-Conv before the end of the discussion period.
>
> > I believe the incorporation of the symmetries discovered by ATLAS leading to parameter efficiency has not been validated. I request the authors to discuss how they ensure effective parameter efficiency with their proposed approach in the rebuttal.
>
> In the third experiment, the gauge equivariant neural network that utilizes our discovered results has less than a third of the parameters compared to the baseline CNN. We have updated the paper to include these counts in Appendix C. In the fourth experiment, Table 3 highlights that the network incorporating our discovered result is roughly seven times more parameter efficient than the baseline.

---

> > ### Author Response · Authors · 2024-11-20
> > **Response to Weakness 5 and Minor Weakness**
> >
> > > How can we randomly sample a localized function $\Phi_c$ from $\Phi$? Do we need to consider the random bias that might be introduced through the random sampling of $\Phi_c$? Besides, is there any potential adversarial impact of the randomly sampled coefficient vector $\eta$ in terms of deriving the group element $g$?
> >
> > We have rechecked our implementation and in a single training step we actually sum the loss over all localized predictors rather than randomly choosing a single predictor. We apologize for the minor error and have updated the text accordingly.
> >
> > The previous text suggests randomly selecting a localized function $\Phi_c$ from the collection of all possible localized functions $\{\Phi_c\}$, not from $\Phi$ itself. Recall that there is a single localized function for each chart. In effect, rather than evaluating the entirety of the loss $\sum_{c} \mathcal{L}(\Phi_{c}(g \cdot x), g \cdot \Phi_{c}(x))$ every single training iteration, we would compute it stochastically. We do not believe this would introduce a significant bias since after enough training iterations, all localized functions will be sampled with roughly equal proportion.
> >
> > The distribution of the coefficient vector $\eta$ may affect the final discovered result. For example, if a system is only equivariant to a subset of a group and the distribution of $\eta$ happens to match this subset, then we may falsely report a symmetry. We found that a normal distribution rarely suffers from this issue and generalizes well across a variety of settings. Thus, it was used for all of our experiments.
> >
> > > How the parameter $\epsilon$ is being selected? An ablation would be great to analyze the robustness of this parameter over the model's performance.
> >
> > In general, $\epsilon$ is chosen so that representatives from the same coset are correctly labeled as duplicates and those from different cosets are labeled as distinct. In many cases, all the representatives of a given coset converge to the exact same value, allowing for a wide range of possible $\epsilon$ consistent with this criteria. This makes it quite easy to tune $\epsilon$ in practice. For instance, in the top-tagging experiment any value from roughly $10^{-4}$ to $10^{-1}$ leads to a final selection of the identity and parity components.

---

> > > ### Author Response · Authors · 2024-11-26
> > > **Comparison with LieGG**
> > >
> > > We have now performed an additional comparison with LieGG by having it search for global equivariances in the PDE experiment. We note that the exact setting of the experiment is slightly different from the one used for ATLAS, but all changes only make discovery easier. In summary, we find LieGG unable to discover a global symmetry with the full details and results available in Appendix D.
> > >
> > > Finally, we note that generalizing LieGG to the case of equivariances when the group acts on $\mathbb{R}^{2}$ is not fully trivial and our implementation is somewhat space consuming. We are open to any suggestions by the reviewer if they know of better generalizations.

---

> ### Author Response · Authors · 2024-12-03
> **A kind reminder**
>
> Dear reviewer,
>
> We sincerely appreciate your detailed feedback and the time you have invested in reviewing our work. We have carefully addressed the points you raised and provided clarifications to any misunderstandings or concerns. If you have additional questions, we would be happy to provide further clarification. If all of your concerns have been addressed, may we kindly request you to increase the score?

---

### Official Review · Reviewer_Jszk · 2024-11-02

**Soundness:** 3
**Presentation:** 2
**Contribution:** 2
**Rating:** 5
**Confidence:** 4

**Summary:**

In this paper, the authors proposed a new method called atlas to capture symmetries in local neighborhoods. Local symmetries of a function is recovered by training local predictor networks and learning a Lie group basis to ensure the learned predictors are equivariant. To illustrate the efficacy of the proposed method, the authors experimented on a few well-designed tasks.

**Strengths:**

1. The problem of local symmetry may have some specific applications as the authors presented in the experiment section. The proposed method might be a good complement to the global sysmmetry discovery methods.

**Weaknesses:**

1. Motivation of this work is not well clarified. Since existing equivariant models can achieve global symmetries, then local symmetries are also guaranteed at the same time. The authors are encouraged to show important problems or areas where we do not care about global symmetries, but insteand local symmetries are more important.

2. Theoretically I am not seeing advantages of the proposed method over existing methods. The authors are encouraged to clarify novelty of this work and present more analysis on why the proposed method is superior to existing methods.

**Questions:**

1. Since Gauge equivariant neural networks could enforce local symmetries, what are the major differences between the proposed atlas and the Gauge equivariant neural networks? Does theorem 1 mean they are equivalent under certain conditions?

2. In Algo.1 the authors train a predictor network for each chart. What are the scales of the chart? Talking about local symmetries, I assume the locality can be referred at different scales. It also depends on the problem/data themselves.

3. What are the trade-offs when we optimize local symmetries using atlas? Do we lose global symmetries? How could we balance these two objectives?

4. Fig. 9, can the authors explain a little bit more on why the proposed method is superior to the LieGAN method? Visually the differences are subtle to me.

---

> ### Author Response · Authors · 2024-11-20
>
> > Motivation of this work is not well clarified. Since existing equivariant models can achieve global symmetries, then local symmetries are also guaranteed at the same time.
>
> The reason why local symmetry must be considered is that local symmetries can be present even in problems that have limited global symmetries. Consider the heat experiment, where the heat source breaks the global symmetry of the system but the local neighborhoods still maintain symmetry. More generally, we believe ATLAS is relevant in a variety of PDE applications.
>
> We also note that discovering a system has a global symmetry is not enough to guarantee it also has local symmetries (e.g. $\frac{\partial u}{\partial t} = x^2 + y^2$ only has a global rotational symmetry).
>
> > Theoretically I am not seeing advantages of the proposed method over existing methods.
>
> ATLAS is superior to alternative symmetry discovery methods in that it is able to discover a class of symmetries that existing methods are unable to: local symmetries. This is concretely demonstrated in the third experiment where we show ATLAS discovers a $\mathrm{SO}(2)$ local rotational symmetry but LieGAN is unable to provide a useful result.
>
> Another novel contribution ATLAS brings is the ability to discover arbitrary discrete symmetries. For instance, in the top-tagging experiment, ATLAS discovers both the parity and identity components of $\mathrm{SO}(1, 3)$. LieGAN, on the other hand, only learns the identity component.
>
> > Since Gauge equivariant neural networks could enforce local symmetries, what are the major differences between the proposed atlas and the Gauge equivariant neural networks? Does theorem 1 mean they are equivalent under certain conditions?
>
> ATLAS is a pipeline that **discovers** local symmetries of certain functions. Gauge equivariant neural networks, on the other hand, are neural networks that **utilize** local symmetries to improve performance when trying to fit a function. At a high level, Theorem 1 states that the symmetries discovered by ATLAS are the same ones that can be used by gauge equivariant neural networks under certain conditions. This allows our discovered result to be used in downstream neural networks. Please let us know if this answers your question.
>
> > In Algo.1 the authors train a predictor network for each chart. What are the scales of the chart? Talking about local symmetries, I assume the locality can be referred at different scales. It also depends on the problem/data themselves.
>
> You are correct that the size of the chart used may depend on the exact problem. For instance, in our experiments the number of charts ranged from $3$ to $19$. With that said, we show in experiment 2 that a single system can display atlas equivariance at multiple scales, suggesting that there is leeway in the size and number of charts to be used for a given task.
>
> For a visual depiction of some of our chart configurations, we refer to Figure 6b and 8. Also, Appendix C contains the exact chart sizes and locations for all experiments.
>
> > What are the trade-offs when we optimize local symmetries using atlas? Do we lose global symmetries? How could we balance these two objectives?
>
> Although ATLAS is primarily designed with local symmetries in mind, we note it can also function as a global discovery pipeline. For instance, in experiment 1 it is used to discover global symmetries and demonstrates competitive results with baselines such as LieGAN. Thus, ATLAS is capable of discovering both local and global symmetries and we do not lose one by trying to discover the other.
>
> > Fig. 9, can the authors explain a little bit more on why the proposed method is superior to the LieGAN method? Visually the differences are subtle to me.
>
> In Figure 9, the transformation produced by ATLAS (middle row) keeps the form of the dataset by rotating each digit of the original datapoint (first row). On the other hand, the transformation produced by LieGAN (last row) clearly distorts the image out of distribution by translating the digits outside of their allotted position. Recall that all elements of the dataset have three digits placed in fixed spots along the equator.
>
> Any proper symmetry must transform one element in the distribution to another one. Hence, LieGAN taking a datapoint out of distribution suggests it is producing a random transformation rather than a true symmetry. Conversely, the transformation suggested by ATLAS keeps the datapoint within the distribution, suggesting that it is a true symmetry.

---

> ### Author Response · Authors · 2024-12-03
> **A kind reminder**
>
> Dear reviewer,
>
> We sincerely appreciate your detailed feedback and the time you have invested in reviewing our work. We have carefully addressed the points you raised and provided clarifications to any misunderstandings or concerns. If you have additional questions, we would be happy to provide further clarification. If all of your concerns have been addressed, may we kindly request you to increase the score?

---

### Official Review · Reviewer_SqyP · 2024-11-03

**Soundness:** 2
**Presentation:** 2
**Contribution:** 1
**Rating:** 3
**Confidence:** 5

**Summary:**

This paper presents ATLAS, a method for discovering local symmetry. It defines atlas locality and atlas equivariance to formalize local symmetry. ATLAS is based on trained predictors, sequentially treating Lie algebra bases and coset representatives as learnable parameters, with the optimization objective being the equivariant loss $\mathcal{L}(\Phi_c(g \cdot x), g \cdot \Phi_c(x))$. It then compares the relationship between gauge equivariance and atlas equivariance. Experimental results demonstrate that this method can correctly identify local symmetry in the task.

**Strengths:**

This paper focuses on the unique problem of local symmetry discovery and formally defines atlas equivariance to distinguish between global symmetry and local symmetry. In addition to Lie algebra bases, ATLAS can learn coset representatives, thereby discovering discrete symmetry.

**Weaknesses:**

- I disagree with the core argument of this paper: previous methods for symmetry discovery could only find global symmetry rather than local symmetry. In Algorithm 1, the atlas $\mathcal{A}$ of the manifold $\mathcal{M}$ needs to be specified manually. However, after completing this step, we can treat the symmetry discovery for each chart $(U_c, \varphi_c)$ as separate subproblems, with their common symmetry being the atlas symmetry. These subproblems do not differ from global symmetry discovery, as mentioned in Definition 2, where each mapping $\Phi_c$ is globally equivariant. Therefore, previous methods for symmetry discovery can be naturally extended to the case of local symmetry. For example, we can apply LieGG [1] on each trained predictor $\Phi_c$, or apply LieGAN [2] on the data of each chart $\mathcal{D_c} = \\{(X_i: U_c \rightarrow \mathbb{R}^{d_{in}}, Y_i: U_c \rightarrow \mathbb{R}^{d_{out}})\\}_{i=1}^{n_c}$.

- ATLAS simply treats group generators as learnable parameters and optimizes the equivariant loss $\mathcal{L}(\Phi_c(g \cdot x), g \cdot \Phi_c(x))$. I think that this approach has limited contributions to the problem of symmetry discovery. It does not show significant advantages over previous methods and has some drawbacks. For example, compared with LieGG [1], which mathematically solves Lie algebra bases, ATLAS has poorer interpretability; compared with LieGAN [2], which directly discovers symmetries from datasets, ATLAS is highly dependent on the accuracy of trained predictors. Furthermore, in Appendix B, it is mentioned that the number of Lie algebra bases $k$ is determined by incrementally increasing from a minimum value. Will this lead to significant computational overhead when the true $k$ is large? If the norm of the weakest generator varies continuously, will changes in the set threshold also affect the final result for $k$?

- The reason why the method can work lacks rigorous explanation. See the first two questions for details.

- Suggestion on presentation: "atlas" refers to a collection of local regions or charts that cover a manifold, while "ATLAS" stands for automatic local symmetry discovery. The use of the same word, differing only in capitalization, may confuse readers.

**References**

[1] Moskalev, Artem, Anna Sepliarskaia, Ivan Sosnovik, and Arnold Smeulders. "Liegg: Studying learned lie group generators." Advances in Neural Information Processing Systems 35 (2022): 25212-25223.

[2] Yang, Jianke, Robin Walters, Nima Dehmamy, and Rose Yu. "Generative adversarial symmetry discovery." In International Conference on Machine Learning, pp. 39488-39508. PMLR, 2023.

**Questions:**

- Does Equation (3) in Section 4.2.1 (calculating the absolute value first and then the cosine similarity) have any practical significance mathematically? I am skeptical about whether it works in all cases. Can you provide a theoretical proof that when $\mathcal{L}_{str}(B)$ is minimized, the number of non-zero elements shared among different basis vectors is also minimized?

- In Section 4.2.2, when learning coset representatives, even though the number of random matrices $K$ is much larger than the actual number of cosets, will all these matrices converge to the strong symmetry embodied by the predictor $\Phi_c$? Formally, if for $\forall C \neq C_1, C_2$, we have $\mathcal{L}(\Phi_c(C_1 \cdot x), C_1 \cdot \Phi_c(x)) < \mathcal{L}(\Phi_c(C_2 \cdot x), C_2 \cdot \Phi_c(x)) \ll \mathcal{L}(\Phi_c(C \cdot x), C \cdot \Phi_c(x))$ (where we omit the normalization notation for brevity). In this case, it is possible that all matrices converge to $C_1$ while missing the symmetry corresponding to $C_2$.

- In Section 5.2, does the $\mathrm{SO}(2)$ transformation refer to rotating the entire field around the origin of the coordinate system, or does it refer to rotating each chart separately around its center point? In Figure 6b, which charts correspond to local predictors $\Phi_c$ that exhibit global symmetry?

- In the assumptions of Theorem 1, it is already mentioned that $M$ is $G$ gauge equivariant. Is it necessary to further elaborate that $M$ is $G$ equivariant in the third paragraph of the proof? When the manifold is Euclidean space, are gauge equivariance and equivariance naturally equivalent?

---

> ### Author Response · Authors · 2024-11-20
> **Response to Weaknesses**
>
> > I disagree with the core argument of this paper: previous methods for symmetry discovery could only find global symmetry rather than local symmetry. [...] Therefore, previous methods for symmetry discovery can be naturally extended to the case of local symmetry.
>
> We clarify that **our main contribution is a novel integration of symmetry discovery techniques within a unique setting as opposed to a fundamentally distinct discovery paradigm.** The leap that our paper takes is the fact that one must consider local symmetry to begin with. Indeed, to the best of our knowledge, applying symmetry discovery methods to localized functions is yet to be studied. While performing this localization may appear to be simple, it is arbitrary without proper theoretical grounding. In our work, this is done by providing a formulation of local symmetry as well as its connection to gauge equivariance to motivate discovery on the localized functions as a meaningful action with practical applications.
>
> We agree in principle that LieGG or LieGAN would be able to discover local symmetries if they are given the individual charts. However, while this is a natural extension of previous methods, it is not a trivial extension as it fundamentally modifies their search space. The point of comparison with global symmetry discovery methods is to show that the hypothesis space of all local symmetries is sometimes more applicable than all global ones.
>
> >  I think that [ATLAS’s] approach has limited contributions to the problem of symmetry discovery. It does not show significant advantages over previous methods and has some drawbacks. For example, compared with LieGG [1], which mathematically solves Lie algebra bases, ATLAS has poorer interpretability; compared with LieGAN [2], which directly discovers symmetries from datasets, ATLAS is highly dependent on the accuracy of trained predictors.
>
> We respectfully disagree with this point. ATLAS relies on an accurate predictor only as much as LieGAN relies on an accurate discriminator. We are even able to discover symmetries when the predictor is imperfect. In the first experiment for example, the predictor used is a naive 5-layer MLP that achieves an 86% test accuracy. Despite this, the discovered invariant metric tensor has a cosine similarity of $0.9996$ with the ground truth. We further note that a predictor brings the advantage that it can be trained independently of the discovery process, unlike the discriminator in the LieGAN approach. With regards to LieGG, the final results of ATLAS are also presented in the form of a Lie algebra basis. Hence, we argue that we have comparable interpretability to LieGG.
>
> Besides, one advancement our paper does provide is the discovery of discrete generators. LieGG and LieGAN would be unable to learn certain discrete generators that our method was shown to recover, such as in the PDE experiment.
>
> > Furthermore, in Appendix B, it is mentioned that the number of Lie algebra bases $k$ is determined by incrementally increasing from a minimum value. Will this lead to significant computational overhead when the true $k$ is large? If the norm of the weakest generator varies continuously, will changes in the set threshold also affect the final result for $k$?
>
> In practice, the dimension of the symmetry group is typically quite small (the highest we found was $7$) so it is computationally feasible to retrain the model $k$ times. We have included a more detailed time complexity analysis in Appendix E of the updated paper. To summarize, the localized predictors will be evaluated $\mathcal{O}(kTP)$ times, where $T$ is the number of training steps and $P$ is the number of predictors. While the regularizations are more expensive in terms of $k$, their constant factor is insignificant compared to the predictors.
>
> By "the norm of the weakest generator varies continuously", we assume you mean that the norm changes at a close to constant rate when $k$ increases. In all of our experiments, we found that the norm of the weakest generator becomes significantly smaller after exceeding the true dimension, so this was not an issue. If we’ve misunderstood your question, please point it out.
>
> > Suggestion on presentation: "atlas" refers to a collection of local regions or charts that cover a manifold, while "ATLAS" stands for automatic local symmetry discovery. The use of the same word, differing only in capitalization, may confuse readers.
>
> Thank you for the comment. We will consider a different name for the model to avoid confusion.

---

> ### Author Response · Authors · 2024-11-20
> **Response to Questions**
>
> > Does Equation (3) in Section 4.2.1 (calculating the absolute value first and then the cosine similarity) have any practical significance mathematically? I am skeptical about whether it works in all cases. Can you provide a theoretical proof that when $\mathcal{L} _ {\text{sbr}}$ is minimized, the number of non-zero elements shared among different basis vectors is also minimized?
>
> To provide theoretical motivation, assume that we have a basis $\\{ b_i \\} _ {i = 1}^{k}$ of some subspace of $\mathbb{R}^{n}$. The number of shared non-zero element pairs is given by $z(b) = \sum_{i = 1}^{k} \sum_{j = i + 1}^{k} \sum_{\ell = 1}^{n} \omega(b_{i, \ell}) \omega(b_{j, \ell})$, where $\omega(x)$ is $1$ if $x \ne 0$ and $0$ otherwise. To construct $z_d$, a differentiable variant of $z$, a natural choice is to replace $\omega(\cdot)$ with $| \cdot |$. From here, $\mathcal{L} _ {\text{sbr}}$ differs from $z_d$ only by a normalization factor in the innermost sum.
>
> In Appendix A of the updated paper, we have included a proof that under certain conditions, $\mathcal{L} _ {\text{sbr}}$ and $z$ are minimized together. These conditions are met by the Lie algebras of many matrix Lie groups, such as $\mathrm{SO}(N)$, $\mathrm{SO}(1, 3)$, and $\mathrm{U}(n)$. Indeed, in the top-tagging experiment the discovered basis has only a single pair of shared non-zero elements when using $\mathcal{L} _ {\text{sbr}}$, but $84$ such pairs using cosine regularization.
>
> You are correct that for arbitrary subspaces the argmins of $z$ and $\mathcal{L} _ {\text{sbr}}$ may disagree (e.g. for $\text{span} \\{  (1, 0.1, 0.1, 0), (-0.9, 0, 0, 1) \\}$. In these cases, $\mathcal{L}_{\text{sbr}}$ trades sparsity in favor of a more orthogonal basis (e.g. $\\{ (1, 0.1, 0.1, 0), (0.0, 0.09, 0.09, 1) \\}$), which is usually desirable.
>
> > In Section 4.2.2, when learning coset representatives, even though the number of random matrices $K$ is much larger than the actual number of cosets, will all these matrices converge to the strong symmetry embodied by the predictor $\Phi_c$? Formally, if for $\forall C  \ne C_1,C _2$, we have $\mathcal{L} (\Phi_c(C_1\cdot x), C_1 \cdot \Phi_c(x))< \mathcal{L}(\Phi_c (C_2 \cdot x), C_2 \cdot \Phi_c(x)) \ll \mathcal{L}(\Phi_c (C⋅x),C⋅\Phi_c(x))$ (where we omit the normalization notation for brevity). In this case, it is possible that all matrices converge to $C_1$ while missing the symmetry corresponding to $C_2$.
>
> We observe an abundance of local minima in the search space, which means it is unlikely gradient descent will take a representative originally near $C_2$ towards $C_1$. For instance, in the special case that $C_1$ and $C_2$ have different determinants, it is improbable that representatives near $C_2$ will converge to $C_1$ since they would have to become singular at some point.
>
> To confirm this more generally, we experiment on the following function: $f(x, y) = \arctan((y + 0.1) / x)$. The identity transformation has zero loss ($10^{-5}$) whereas a rotation by $180$ degrees has minor loss ($0.16$). The full results are available in Appendix D, but in summary we discover an average of $8$ representatives from the rotated component in the top $24$ unfiltered cosets.
>
> If identity collapse does become an issue, one can add a regularization term to encourage distinct coset representatives (e.g. cosine similarity).
>
> > In Section 5.2, does the $\mathrm{SO}(2)$ transformation refer to rotating the entire field around the origin of the coordinate system, or does it refer to rotating each chart separately around its center point? In Figure 6b, which charts correspond to local predictors $\Phi_c$ that exhibit global symmetry?
>
> The rotation is done separately around each chart’s center point. Specifically, transformations of the individual charts are applied with respect to the origin of the local coordinate system as defined by $\varphi_c$. In the atlases that we use, the origin of the local coordinate system is chosen to be the center of the chart.
>
> In Figure 6b, we have a local predictor $\Phi_c$ for every single chart. Our discovered result corresponds to the global symmetry that all $\Phi_c$ simultaneously share. Hence, the local predictor for every chart demonstrates the $\mathrm{SO}(2)$ symmetry (though to possibly varying extents).
>
> > In the assumptions of Theorem 1, it is already mentioned that M is G gauge equivariant. Is it necessary to further elaborate that M is G equivariant in the third paragraph of the proof? When the manifold is Euclidean space, are gauge equivariance and equivariance naturally equivalent?
>
> In Euclidean space, while it is true that a $G$ gauge equivariant CNN is also $G$ equivariant, gauge equivariance does not necessarily imply global equivariance. For instance, if we ease the restriction of spatial weight sharing in a CNN (thereby allowing each kernel to also depend on location), then the network may be gauge equivariant but not globally equivariant.

---

> > ### Comment · Reviewer_SqyP · 2024-11-26
> >
> > Thank you for your reply. I will maintain my score because my core concerns still remain, as detailed below.
> >
> > - I acknowledge the contribution of this paper in applying symmetry discovery to local functions. However, I believe that merely introducing a new problem setting or application scenario, without methodological innovation, is insufficient for publication in a conference.
> >
> > - It remains unclear why ATLAS is more specialized than previous methods in the problem of local symmetry discovery. Although ATLAS and LieGAN each have their trade-offs (in terms of predictor and discriminator training), ATLAS does not appear to have a significant advantage over LieGG.

---

> > > ### Author Response · Authors · 2024-11-27
> > >
> > > We thank the reviewer for their feedback and thoughtful response. To provide clarification for further readers, we briefly address a few of the discussion points.
> > >
> > > > However, I believe that merely introducing a new problem setting or application scenario, without methodological innovation, is insufficient for publication in a conference.
> > >
> > > We argue that we demonstrate methodological innovation through our ability to discover arbitrary discrete generators. This is something that baselines such as LieGG and LieGAN are unable to achieve.
> > >
> > > > ATLAS does not appear to have a significant advantage over LieGG.
> > >
> > > While LieGG can only discover invariances of functions, ATLAS can readily search for equivariances. Although the former may be generalized to consider equivariances, this induces a significant computational cost in our context since the polarization matrix scales linearly with the total number of output pixels in the dataset (see Appendix D for details).

---

### Official Review · Reviewer_cuZV · 2024-11-04

**Soundness:** 3
**Presentation:** 2
**Contribution:** 2
**Rating:** 6
**Confidence:** 2

**Summary:**

This work proposed ATLAS to recover local symmetries (both continuous and discrete) from a dataset. By incorporating the symmetries discovered by ATLAS in a gauge equivariant CNN can lead to better performance and parameter efficiency.

**Strengths:**

1. The figures provided in the paper are of high quality, easy to understand

2. The algorithm proposed is straightforward and concise.

**Weaknesses:**

1. The motivation for discovering local symmetries through ATLAS is not clearly articulated. The current description is somewhat abstract, which may make it challenging for general readers to understand. It would be helpful to clarify what the symmetries represent in the context of the three experiments. Are these symmetries intended primarily to enhance performance in gauge-equivariant CNNs, or do they have a broader purpose?

2. While the three experiments in Section 5 provide concrete examples, the connections between group actions, cosets, and the datasets are not immediately apparent. It would be beneficial for the authors to explicitly state these connections within each experimental subsection to enhance clarity.

3. The objective of the experimental section lacks clarity. If the primary goal is benchmarking, the comparisons with other methods appear limited. On the other hand, if the goal is to illustrate the advantages of local symmetries over global symmetries, it remains unclear how these local symmetries contribute to downstream tasks. Providing more specific insights into the role and utility of local symmetries in these tasks would improve the section’s effectiveness.

**Questions:**

See weakness

---

> ### Author Response · Authors · 2024-11-20
>
> > The motivation for discovering local symmetries through ATLAS is not clearly articulated.
>
> The reason why local symmetry must be considered is that local symmetries can be present even in problems that have limited global symmetries. You are correct that incorporating local symmetries into a gauge equivariant CNN is one practical application. In general, however, learning the local symmetries of a function also gives greater insight into its behavior and properties. Existing discovery pipelines are unable to realize these useful symmetries, giving the motivation for ATLAS.
>
> In our experiments, symmetries represent the transformations we can make on local neighborhoods of the input feature field that, after applying the task function, will result in a corresponding change in the output field. For instance, in the heat experiment, the discovered result of $\mathrm{O}(2)$ indicates that if we rotate or reflect a local neighborhood of the temperature field, the corresponding output neighborhood will be rotated or reflected in the exact same manner after we simulate heat flow
>
> > While the three experiments in Section 5 provide concrete examples, the connections between group actions, cosets, and the datasets are not immediately apparent. It would be beneficial for the authors to explicitly state these connections within each experimental subsection to enhance clarity.
>
> We believe that we have included the ground truth symmetry for each dataset in the main experiment description. We also attempt to cover the meaning of our results in the problem. For instance, in the top-tagging experiment, Figure $4$ explains the role of each infinitesimal generator and Figure $5$ provides context for the two discovered cosets. We kindly ask the reviewer to point out any additional detail that should be clarified.
>
> > The objective of the experimental section lacks clarity. If the primary goal is benchmarking, the comparisons with other methods appear limited. On the other hand, if the goal is to illustrate the advantages of local symmetries over global symmetries, it remains unclear how these local symmetries contribute to downstream tasks. Providing more specific insights into the role and utility of local symmetries in these tasks would improve the section’s effectiveness.
>
> We will clarify the role of each experiment. Our first experiment aims to prove the capability of ATLAS as a symmetry discovery tool. The second shows its resilience to the choice of atlas used. The third demonstrates that ATLAS can discover local symmetries in cases where LieGAN cannot find global symmetries. The fourth highlights the ability of ATLAS to discover local symmetry that is useful in downstream prediction tasks in real-world datasets.
>
> Mainly, we want to show that ATLAS is capable of discovering local symmetries. The tests of downstream models are to further verify the correctness of the learned symmetries as well as demonstrate their practical applications.

---

> > ### Author Response · Authors · 2024-12-03
> > **A Kind Reminder**
> >
> > Dear reviewer,
> >
> > We sincerely appreciate your detailed feedback and the time you have invested in reviewing our work. We have carefully addressed the points you raised and provided clarifications to any misunderstandings or concerns. If you have additional questions, we would be happy to provide further clarification. If all of your concerns have been addressed, may we kindly request you to increase the score?

---

### Official Review · Reviewer_H7L3 · 2024-11-04

**Soundness:** 3
**Presentation:** 2
**Contribution:** 3
**Rating:** 6
**Confidence:** 3

**Summary:**

The paper presents ATLAS (Automatic Local Symmetry Discovery), an approach for discovering local symmetries in various data forms.  The authors provide formal descriptions of the locality and equivariance of ATLAS, along with algorithms for discovering infinitesimal generators and discrete symmetries. Results from these experiments show that integrating the discovered symmetries into downstream models can enhance performance. Through experiments involving top-quark tagging, synthetic PDEs, MNIST classification, and climate segmentation, the authors demonstrate the effectiveness of ATLAS in discovering local symmetries and integrating them into gauge equivariant networks, leading to improved performance metrics.

**Strengths:**

1. The paper is well-motivated, presenting a new conceptual framework. The authors validate their approach across multiple tasks and demonstrate its effectiveness.

**Weaknesses:**

1. The definitions of Atlas Locality and Atlas Equivariance both contain the term "Atlas," an abbreviation for the authors' method, i.e., Automatic Local Symmetry Discovery. Defining a mathematical object using the abbreviation of a specific method is uncommon and could lead to confusion. Additionally, the authors do not clearly differentiate atlas locality from general locality, nor do they clarify how atlas equivariance differs from equivariance to conventional local transformations.
2. While the authors emphasize the importance of local equivariance, many deep learning tasks, such as natural language processing and video understanding, exhibit long-range dependencies, highlighting the coupling of local and global equivariance. Is the proposed framework applicable in these scenarios?
3. The authors report only the final results of their complete method in each experiment, lacking ablation studies and fine-grained visualizations of features to provide a more credible explanation of their approach.
4. The descriptions of experimental details are somewhat lacking, particularly concerning the discovery of symmetry and its integration with neural networks on specific datasets.
5. The paper conducts many experiments, and for the MNIST classification task, it projects the data onto a spherical space. What are the physical meanings or practical applications of this projection?
6. Although the paper provides the steps of the ATLAS algorithm, there is limited analysis on the time and space complexity of the algorithm. It is recommended to include this section to assess its feasibility for applications on large-scale datasets.

**Questions:**

see Weaknesses in the previous section

---

> ### Author Response · Authors · 2024-11-20
> **Response to Weakness 1-4**
>
> > The definitions of Atlas Locality and Atlas Equivariance both contain the term "Atlas," an abbreviation for the authors' method, i.e., Automatic Local Symmetry Discovery. Defining a mathematical object using the abbreviation of a specific method is uncommon and could lead to confusion.
>
> To clarify, we did not "define mathematical objects using the abbreviation of our method’’. The terms atlas local and atlas equivariance take their name from atlases, an established mathematical term for a collection of charts (see section 3). We understand this can be confusing and will change the model name away from ATLAS to avoid any ambiguities.
>
> > Additionally, the authors do not clearly differentiate atlas locality from general locality, nor do they clarify how atlas equivariance differs from equivariance to conventional local transformations.
>
> We are not sure what the reviewer means by “general locality” and “conventional local transformations”, since the meaning of locality heavily depends on context. If the reviewer could clarify these concepts, we would be happy to discuss their connections with atlas locality and atlas equivariance.
>
> > While the authors emphasize the importance of local equivariance, many deep learning tasks, such as natural language processing and video understanding, exhibit long-range dependencies, highlighting the coupling of local and global equivariance. Is the proposed framework applicable in these scenarios?
>
> Our framework is partially applicable in the scenarios. At a high level, there is a semantic connection between the ideas of local symmetry and local-global coupling in that they both deal with interactions at various scales. Our work may help with video understanding since we can discover local transformations in a frame that correlate with a similar transformation in the next one. More generally, ATLAS may be applied to many time-series problems on arbitrary manifolds.
>
> Since our work focuses on maps between feature fields, we do not immediately see a direct application in natural language processing, but are open to any suggestions.
>
> > The authors report only the final results of their complete method in each experiment, lacking ablation studies and fine-grained visualizations of features to provide a more credible explanation of their approach.
>
> We have included an ablation study on the choice of atlases, but agree that more ablation studies on other factors would be helpful.
>
> In the second experiment, we test the sensitivity of our method to the choice of atlas by running the discovery process on two different atlases. We concluded that our method has significant leeway with respect to the exact charts used.
> **We have now performed two additional ablations.** The full results are available in Appendix D of the revised paper, with a summary provided below.
>
> First, we test the effectiveness of the standard basis regularization by replacing it with cosine similarity in the top-tagging experiment. When using the standard basis regularization, the resultant basis has only a single pair of generators that share non-zero elements. If we instead use cosine similarity we observe all $21$ pairs sharing non-zero elements, emphasizing the effectiveness of our regularization to bring the basis into standard form.
>
> Also, we retry the heat experiment without normalizing the cosets during the discovery process. Without normalization, our method is usually able to recover representatives of the identity and reflection component but the representatives of each coset have greater variance and there are more irrelevant cosets. Therefore, the filtration process occasionally would fail and falsely report more than two cosets. This suggests the normalization step is helpful to coset discovery.
>
> > The descriptions of experimental details are somewhat lacking, particularly concerning the discovery of symmetry and its integration with neural networks on specific datasets.
>
> For detailed steps about how to integrate symmetries into a gauge equivariant CNN, we refer to [1].
>
> For symmetry discovery, we try to include important information, such as the number of generators and cosets, in the experiment details and leave the chart sizes and less consequential hyperparameters in the appendix for brevity. We kindly ask the reviewer to point out any specific details they feel should be clarified.
>
> **References:**
>
> [1]: Taco Cohen, Maurice Weiler, Berkay Kicanaoglu, and Max Welling. Gauge equivariant convolutional networks and the icosahedral cnn. In International conference on Machine learning, pp. 1321–1330. PMLR, 2019.

---

> > ### Author Response · Authors · 2024-11-20
> > **Response to Weakness 5-6**
> >
> > > The paper conducts many experiments, and for the MNIST classification task, it projects the data onto a spherical space. What are the physical meanings or practical applications of this projection?
> >
> > The MNIST experiment is a toy example to verify our method.  It simulates a scenario where we are asked to classify different objects on a curved manifold. We can imagine a real-world application when the manifold is the Earth and the goal is to identify different events across the surface, as was done with weather patterns in the fourth experiment.
> >
> > > Although the paper provides the steps of the ATLAS algorithm, there is limited analysis on the time and space complexity of the algorithm. It is recommended to include this section to assess its feasibility for applications on large-scale datasets.
> >
> > Thank you for the suggestion. In Appendix E of the updated paper, we have included a more thorough analysis of the time and space complexity in terms of the dataset and hyperparameters such as the dimensionality and the number of cosets in the group.
> >
> > In summary, letting $P$ denote the number of predictors and $T$ the number of training steps in a given run, the space complexity is given by $\mathcal{O}(k + P)$ for the infinitesimal generator discovery and $\mathcal{O}(K + P)$ for the discrete discovery. The time complexity is $\mathcal{O}(k^2T(k + P))$ for the infinitesimal generator discovery and $\mathcal{O}(K(TP + q))$ for the discrete discovery.

---

> > > ### Comment · Reviewer_H7L3 · 2024-11-26
> > >
> > > Thank you for your response. My concern has been addressed, and I will maintain my original rating of 6.

---

### Author Response · Authors · 2024-11-20
**General Response to the Reviewers**

We appreciate the feedback and detailed comments left by the reviewers. We are encouraged that reviewers find local symmetry discovery to be a novel problem (R3) with practical applications (R4). We are also pleased that our paper is considered well-motivated (R1,5) with proper presentation (R2,5) and demonstrates effective results across various domains (R1,5).


We have revised the paper to address some of the questions posed by reviewers. The major changes include an analysis of the time and space complexity of ATLAS, a comparison to LieGG, additional experiments as well as ablations, and a proof regarding the minimization of the standard basis loss. These changes can primarily be found in the Appendix and are highlighted in red.

---

### Meta-Review · Area_Chair_EisW · 2024-12-19

**Metareview:**

This paper presents an automatic local symmetry discovery method. Experiments on several tasks demonstrate the effectiveness of the proposed method. After the rebuttal, it receives mixed ratings, including three borderline accept with relatively low confidence, one borderline reject with high confidence, and one reject with high confidence. The advantages, including the clear motivation, straightforward and concise algorithm, and good results,  are recognized by the reviewers. However, there are still several concerns that need to be addressed, including the insufficient contribution, unconvincing analysis about the improvement and theoretical advantages, and so on. I think the current manuscript does not meet the requirements of this top conference. I suggest the authors carefully revise the paper and submit it to another relevant venue.

**Additional Comments On Reviewer Discussion:**

The response well address some concerns. However, two reviewers still think the current paper does not meet the requirements of this top conference with high confidence. I agree with them.

---

### Decision · Program_Chairs · 2025-01-22

Reject